# Boundary cells restrict dystroglycan trafficking to control basement membrane sliding during tissue remodeling

Shelly TH McClatchey[1†], Zheng Wang[2,3,4,5†], Lara M Linden[1], Eric L Hastie[1], Lin Wang[2,3], Wanqing Shen[2,3], Alan Chen[1], Qiuyi Chi[1], David R Sherwood[1*]

[1]Department of Biology, Duke University, Durham, United States; [2]Center for Tissue Engineering and Regenerative Medicine, Union Hospital, Wuhan, China; [3]Tongji Medical College, Huazhong University of Science and Technology, Wuhan, China; [4]Department of Gastrointestinal Surgery, Union Hospital, Wuhan, China; [5]Development and Molecular Oncology Laboratory, Union Hospital, Wuhan, China

**Abstract** Epithelial cells and their underlying basement membranes (BMs) slide along each other to renew epithelia, shape organs, and enlarge BM openings. How BM sliding is controlled, however, is poorly understood. Using genetic and live cell imaging approaches during uterine-vulval attachment in *C. elegans*, we have discovered that the invasive uterine anchor cell activates Notch signaling in neighboring uterine cells at the boundary of the BM gap through which it invades to promote BM sliding. Through an RNAi screen, we found that Notch activation upregulates expression of *ctg-1*, which encodes a Sec14-GOLD protein, a member of the Sec14 phosphatidylinositol-transfer protein superfamily that is implicated in vesicle trafficking. Through photobleaching, targeted knockdown, and cell-specific rescue, our results suggest that CTG-1 restricts BM adhesion receptor DGN-1 (dystroglycan) trafficking to the cell-BM interface, which promotes BM sliding. Together, these studies reveal a new morphogenetic signaling pathway that controls BM sliding to remodel tissues.

*For correspondence: david.sherwood@duke.edu

†These authors contributed equally to this work

## Introduction

The basement membrane (BM) is a cell-associated, dense, sheet-like form of extracellular matrix that underlies all epithelia and endothelial tissue, and surrounds muscle, fat, and Schwann cells (*Halfter et al., 2015*; *Yurchenco, 2011*). BMs are built on polymeric laminin and type IV collagen networks that arose at the time of animal multicellularity, and may have been required for the evolution of complex tissues (*Hynes, 2012*; *Ozbek et al., 2010*). Consistent with this idea, BMs provide tissues with mechanical support, barrier functions, and cues for polarization, differentiation and growth (*Breitkreutz et al., 2013*; *Hay, 1981*; *Poschl et al., 2004*; *Rasmussen et al., 2012*; *Suh and Miner, 2013*; *Yurchenco, 2011*).

Although it was generally thought that cell-BM interactions are static, live imaging studies have revealed that cell-BM interfaces are highly dynamic (*Morrissey and Sherwood, 2015*). One of the most dramatic examples of this mobility is cell-BM sliding, during which epithelial cell layers and their underlying BM sheets move (slide) along one another independently to regulate tissue remodeling or renewal. Examples of cell-BM sliding are varied and include egg chamber rotation in *Drosophila*, where the follicle cells move along BM and deposit constricting strips of aligned type IV collagen, which direct egg elongation along the anterior-posterior axis (*Cetera and Horne-*

**eLife digest** All tissues in the human body are encased with a thin, dense, web of proteins called the basement membrane. These membranes separate and help shape tissues, while protecting them from mechanical damage such as stretching. However, despite serving as important barriers, basement membranes must be rapidly removed at specific times and places to allow tissues to grow and change shape. This process also permits cells to enter and leave tissues, such as immune cells when they search for disease-producing agents. This creates a paradox: how does a tissue create a tough, impassable barrier that can also be rapidly removed at specific times and places?

Recent research has shown that basement membranes can slide over a tissue, and thus rapidly move the barriers to create openings for tissue growth and reshaping. However, it was not known how the cells within tissues that are normally firmly attached to basement membrane control when and where basement membrane can slide.

Now, McClatchey, Wang et al. have carefully observed the tissues and basement membranes in a small worm called *Caenorhabditis elegans* under a microscope. This revealed that a single cell, called the anchor cell, relays a signal that instructs a group of neighboring cells to let go of the basement membrane at a specific time to allow tissue reshaping. Further experiments revealed that this signal causes cells to reduce the amount of a protein named dystroglycan at their surface.

Dystroglycan is present in most tissues and helps stick the cells of tissues to basement membranes. The loss of dystroglycan was previously reported to promote the spread of cancer, although its role in cancer progression was not clear. The findings of McClatchey, Wang et al. now suggest that tumors that lose dystroglycan might allow the basement membranes surrounding them to slide, creating openings that allow the cancers to spread.

Finally, McClatchey, Wang et al. also found that a protein named CTG-1, one of a family of proteins thought to regulate the movement of proteins within cells, restricts the levels of dystroglycan at cell surface. As such, the next challenge will be to understand exactly how CTG-1 limits the amount of dystroglycan at the cell surface.

*Badovinac, 2015*; *Haigo and Bilder, 2011*). During salivary gland growth in vertebrates, the BM slides away from the bud tip toward the duct, allowing bud expansion while restricting growth at the duct (*Harunaga et al., 2014*). Further, BM labeling and pulse chase experiments revealed that intestinal epithelial cells derived from the stem cell crypt slide along the BM towards the villus tips during differentiation to rapidly renew the gut epithelium (*Clevers, 2013*; *Trier et al., 1990*). Cell-BM sliding may be a common morphogenetic process that regulates organ shaping during development, tissue homeostasis, and diseases such as cancer, where remodeling of cell-BM interfaces frequently occur (*Kelley et al., 2014*; *Rowe and Weiss, 2008*). Because of the challenge of visualizing and experimentally examining dynamic cell-BM interactions in vivo, however, mechanisms controlling cell-BM sliding remain largely unknown.

An experimentally tractable model to examine cell-BM sliding during tissue remodeling is uterine-vulval attachment in *C. elegans* (*Schindler and Sherwood, 2013*), a developmental process that is necessary for effective mating and egg laying in the worm. During the mid-L3 larval stage, the uterine-vulval connection is initiated by a specialized uterine cell, the anchor cell (AC), that breaches the BM that separate these tissues and attaches to the underlying vulval cells. Following AC invasion, the gap in the BM widens further, which allows additional connection between uterine and vulval cells (*Ihara et al., 2011*). BM gap widening does not involve BM degradation. Instead, optical highlighting of BM and manipulation of tissue dynamics has shown that growth and morphogenesis of the uterine and vulval tissues generate forces on the BM that drive BM sliding over the vulval and uterine cells to further expand the gap (*Ihara et al., 2011*). The vulval cells have a key role in controlling the extent of BM movement. The centrally located vulval E and F cells, which contact the BM gap boundary, undergo precisely timed divisions to initiate BM sliding–cell rounding during divisions dramatically reduces cell contact with the BM and allows the BM to slide over the vulE and F cells (*Matus et al., 2014*). The BM stops sliding on the non-dividing vulD cell, which concentrates the BM

adhesion receptor INA-1/PAT-3 (integrin) to stabilize the BM gap boundary (*Ihara et al., 2011*; *Matus et al., 2014*). While vulval cell divisions control BM sliding on the vulval side of the uterine-vulval connection, the role of the uterine π cells, which flank the AC, and sit on the opposing side of the BM gap boundary, remain unclear.

Many receptors bind BM components and are possible regulators of BM sliding. Two of the most prominent classes of adhesion receptors that link BM to the cytoskeleton are integrin family members and the receptor dystroglycan (*Bello et al., 2015*; *Kramer, 2005*; *Yurchenco, 2011*). Most studies have focused on how these receptors are activated or upregulated to strengthen adhesion; however there is a growing appreciation of the importance of integrin and dystroglycan downregulation in morphogenetic, homeostatic, and disease processes (*Agrawal et al., 2006*; *Bouvard et al., 2013*; *Miller et al., 2015*; *Nakaya et al., 2013*). Cells utilize a variety of mechanisms to reduce BM receptor adhesion. These include transcriptional downregulation, production of negative regulators that interfere with receptor activation, alterations in phosphorylation status, and changes in localization and trafficking (*Bouvard et al., 2013*; *Nakaya et al., 2013*; *Poulton and Deng, 2006*). Whether these strategies are used to modulate adhesion to control BM sliding is unknown.

To address how the uterine BM gap boundary cells control BM sliding during uterine-vulval attachment in *C. elegans*, we carried out a mutagenesis screen. Through this screen we identified a putative null mutant in the gene encoding the LIN-29 protein, which is a Kruppel-family EGR (early growth response) protein (*Harris and Horvitz, 2011*), that is deficient in BM sliding. We show that LIN-29 is expressed and functions in the invading AC to promote BM sliding. We further find that LIN-29 regulates BM sliding through its role in maintaining AC expression of LAG-2, a transmembrane Notch ligand, which activates Notch signaling in the neighboring BM gap boundary cells, the uterine π cells that sit on BM next to the nascent breach. Through a targeted uterine-specific RNAi screen of putative direct Notch targets, we find that Notch activation in these uterine cells upregulates *ctg-1* expression, which encodes a Sec14-GOLD domain phosphatidylinositol transfer protein that belongs to a class of proteins implicated in regulating vesicle trafficking (*Grabon et al., 2015*). Finally, using photobleaching, CRISPR/Cas-9 targeted knockout, and cell-specific rescue experiments, we demonstrate that CTG-1 functions in the uterine π cells to limit the cell surface trafficking of DGN-1 (dystroglycan), and that reduction of DGN-1 is sufficient to promote BM sliding. Together these studies identify a new morphogenetic pathway—from signaling mechanism to effector—that promotes BM sliding, which can be used by invasive cells to enlarge BM openings and allow the direct interaction of cells between tissues.

## Results

### Summary of AC invasion and BM gap expansion

During larval development, the uterine and vulval tissues in *C. elegans* are initially separated by the juxtaposed gonadal and ventral BMs. A specialized uterine cell, the AC, breaches these BMs during the mid-L3 larval stage to initiate uterine-vulval connection (*Sherwood and Sternberg, 2003*). AC invasion and BM remodeling occur in tight coordination with vulval development and can be staged according to the centrally located 1° fated P6.p vulval precursor cell (VPC) divisions. During invasion (P6.p 2 and 4-cell stages) the gonadal and ventral BMs are breached and fuse at the edges of the invading AC (*Figure 1A*, *Ihara et al., 2011*). Following AC invasion, the vulval cells grow, invaginate, and divide (*Ihara et al., 2011*). The rapid expansion and invagination of the vulval cells generate forces on the BM that promote expansion of the nascent BM opening through BM sliding over the vulval and uterine cells that sit at the boundary of the BM gap. Expansion of the BM gap allows direct contact between the cells that mediate uterine-vulval attachment (*Ihara et al., 2011*). Precisely timed divisions of the BM gap boundary vulval P6.p descendants (the vulE and vulF cells) during the late L3 stage reduce BM adhesion, allowing the BM to slide over these dividing vulval cells (*Matus et al., 2014*). The BM gap halts its expansion over the non-dividing vulD cells during the early L4 stage (P6.p 8-cell stage); the vulD cells further stabilize the BM gap boundary by upregulating the integrin heterodimer INA-1/PAT-3 and the adhesion regulator VAB-19 (the worm ortholog of Kank) (see *Figure 1A*, *Ihara et al., 2011*; *Matus et al., 2014*). The ventral uterine (VU) cells neighboring the AC are also initially in contact with the BM. During the time of AC invasion, the adjacent six VU cells are specified to adopt a π fate through LAG-2/LIN-12 (Notch) signaling by the AC.

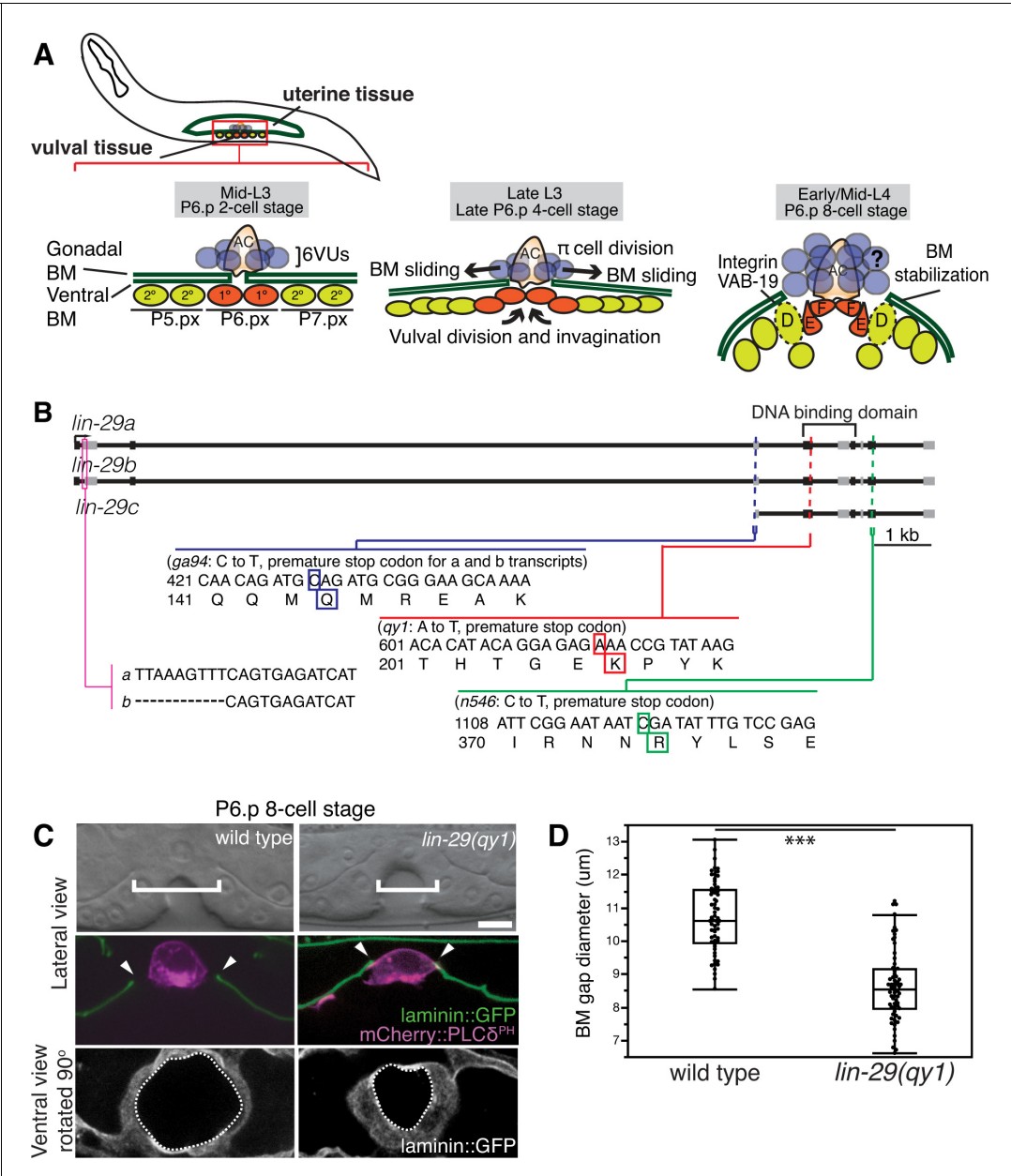

**Figure 1.** BM sliding during uterine-vulval connection and isolation of *lin-29(qy1)*. (**A**) Top left: During the mid-L3 larval stage (vulval P6.p 2-cell stage), the uterine anchor cell (AC) breaches the juxtaposed gonadal and ventral BMs (green) and contacts the central 1° fated vulval precursor cells (VPCs; P6. px cells, red). Middle: After AC invasion is completed at the late L3 (P6.p 4-cell stage), VPC growth, divisions, and invagination drive BM sliding, which widens the BM gap. Ventral uterine (VU) π cells (blue) also begin to divide at the late L3. Right: By the early-to-mid L4 (P6.p 8-cell stage), the BM stops sliding and is stabilized over the non-dividing 2° fated vulD cells (dotted line) by vulD expression of cytoplasmic VAB-19 (Kank) and cell surface integrin. BM sliding allows direct connection between vulE and F cells with uterine π cells, which form the mature uterine-vulval connection. The regulation of BM sliding by the uterine π cells is unknown (represented with '**?**'). (**B**) A schematic diagram of the *lin-29* gene. Allele *qy1* (red) creates an early stop codon affecting all three transcripts. Allele *ga94* (blue) affects the *lin-29a/b* transcripts but leaves the translation of *lin-29c* largely unaffected, while *n546* (green) introduces a nonsense mutation affecting all three encoded isoforms of the LIN-29 protein. (**C**) A lateral, central plane (with the AC in focus) DIC image (top) with BM (laminin::GFP) and AC (*cdh-3 > mCherry::PLCδ^{PH}*) fluorescence (middle) and ventral view of laminin::GFP (bottom) showing the circular opening of the BM at the uterine-vulval connection at the P6.p 8-cell stage. In *lin-29(qy1)* mutants the BM gap (top, white brackets=diameter; middle, arrowheads=edges of gap; bottom, white dashed lines=outline of circular BM gap) failed to widen after AC invasion. Scale Bar, 5 μm. (**D**) Quantification of the BM gap diameter at the P6.p 8-cell stage in wild type (n = 66) and *lin-29(qy1)* mutants (n = 65). Box signifies first and third quartiles and median measurement; whisker ends signify minimum and maximum values that fall within 1.5 times the interquartile range. The asterisks denote a statistically significant difference (*** indicates p<0.001 Wilcoxon rank sum test).

*Figure 1 continued on next page*

*Figure 1 continued*

The following source data is available for figure 1:

**Source data 1.** BM gap diameter in wild type vs. *lin-29(qy1)*.

During BM sliding the uterine π cells undergo one round of division (*Newman, 1995*). The role of the π cells, the uterine BM gap boundary cells, in regulating BM sliding is unknown.

## The Kruppel-family early growth response (EGR) gene *lin-29* promotes BM sliding

To identify genes that regulate the morphogenetic process of BM sliding, we performed an F2 forward mutagenesis genetic screen (*Jorgensen and Mango, 2002*; see Experimental Procedures). We screened the F2 progeny of 12,000 mutagenized F1 animals and selected fertile mutants with a protruding vulva (Pvl) phenotype, which often results from defects in uterine-vulval connection. Isolated Pvl mutants were then examined at high magnification (1000x) using differential interference contrast (DIC) microscopy at the early L4 stage when BM position can be visualized as a phase dense line separating the uterine and vulval tissue. Through this screen we identified 10 mutants with defects in BM remodeling during uterine-vulval attachment (see Materials and methods). We focused on the mutant *qy1*, which had a highly penetrant defect in the BM gap expansion after the AC invasion (see below).

To determine the molecular nature of the *qy1* mutation, we used a single-nucleotide-polymorphism (SNP) based mapping strategy and complementation analysis (see Experimental Procedures; *Davis et al., 2005*). Using this strategy, we mapped the causative mutation of the *qy1* allele to *lin-29*, which encodes a Kruppel-family EGR zinc-finger protein transcription factor (*Harris and Horvitz, 2011*; *Rougvie and Ambros, 1995*). We sequenced the *qy1* allele and found that it introduces a nonsense mutation near the start of the region encoding the DNA binding domain in all three known isoforms of the gene (*Figure 1B*). As this disrupts translation prior to the previously characterized null allele *lin-29(n546)* (*Rougvie and Ambros, 1995*), it suggests that *qy1* is a null allele of the *lin-29* gene.

To better visualize and precisely quantify the BM gap defect after the loss of *lin-29*, we crossed integrated transgenes expressing a functional fusion of the main structural BM component laminin to GFP (laminin::GFP) and an AC membrane marker (*cdh-3* > mCherry::PLCδ$^{PH}$) into *lin-29(qy1)* mutant animals (*Figure 1C,D*). Laminin::GFP shows the same localization as immunolocalized laminin (*Sherwood and Sternberg, 2003*). To quantify the BM gap defect, we measured the diameter of the BM gap at the early L4 (P6.p 8-cell stage) in the plane where the AC is in focus in wild type and *qy1* worms (*Figure 1D*). The gap in the BM was significantly reduced in *qy1* mutant animals (*Figure 1C,D*). Further, we found that that while in wild type animals, the BM moved away from both the anterior and posterior sides of the AC in most animals (n = 13/20 animals; the remaining 7 moved on one side), in *qy1* animals the BM never moved away from both sides of the AC (n = 0/20 animals) and only occasionally moved slightly away from one side (n = 5/20 animals).

We postulated that the defect in BM position in *lin-29* mutant animals could result from the absence of BM sliding or alternatively the inappropriate deposition of new BM after sliding. To differentiate between these possibilities we directly examined BM sliding by performing optical highlighting experiments using transgenic animals expressing laminin tagged with the photoconvertible fluorophore Dendra (laminin::Dendra; *Figure 2A*, *Ihara et al., 2011*). Dendra is a stable, photoconvertible fluorescent protein that irreversibly switches from green to red following exposure to short wavelength light (*Gurskaya et al., 2006*). Segments of the BM adjacent to the invading AC were photoconverted at the P6.p 4-cell stage and imaged at the P6.p 8-cell stage. While marked edges of the BM gap moved away from the AC and rested over the vulD cell in wild type animals (n = 10/10 animals), optically highlighted segments maintained contact with the AC in *lin-29(qy1)* mutants and failed to move over the vulD cell (n = 18/18 animals; *Figure 2A*). These observations indicate that the BM in *lin-29* mutant animals fails to slide away from the AC and excludes additional BM deposition as a mechanism for the defect. Notably, the morphology of the developing vulva—including the number of VPCs and their relative positions (n = 15/15 animals, *Figure 2B*) and vulval

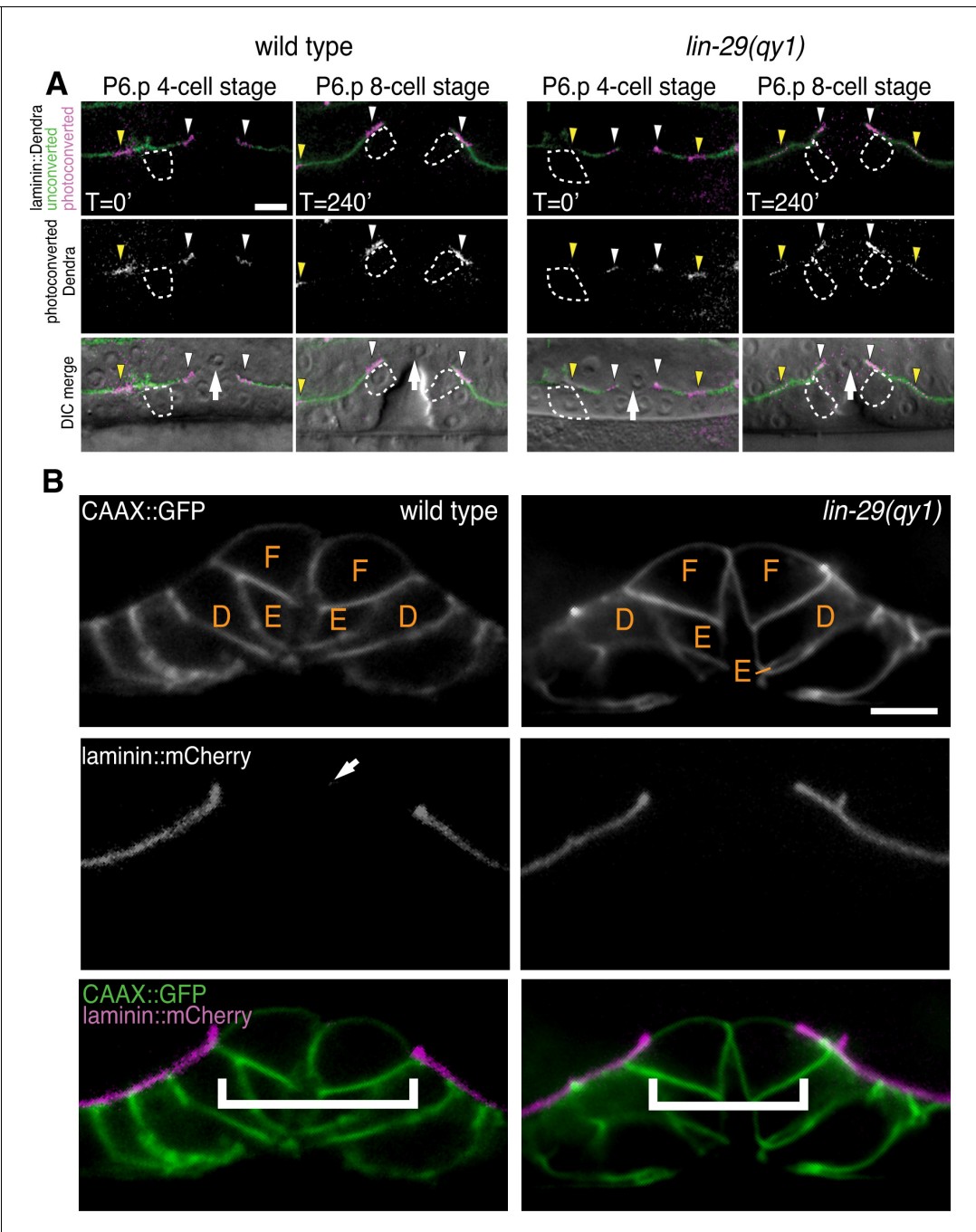

**Figure 2.** *lin-29(qy1)* mutants are defective in BM sliding but have a normal vulval precursor morphology. (**A**) Fluorescence overlays of optical highlighting (photoconversion) of ~5 μm long laminin::Dendra segments (magenta in top; white in middle; white arrowheads highlighted regions next to BM gap, yellow arrowheads highlighted regions farther from gap) in wild type (left) versus *lin-29(qy1)* mutant (right). The fate of optically highlighted BM at the P6.p 4-cell stage (T=0') was assessed at the P6.p 8-cell stage (T=240'). In wild type animals, optically highlighted laminin::Dendra at the edges of the BM gap slid from a position over the vulF precursor cell to over the vulD cell (indicated with dashed outline; n = 10/10 animals). In contrast, in *lin-29(qy1)* mutants the optically highlighted laminin did not move over the vulD cell (n = 18/18 animals). The arrow in the bottom panels indicates the AC. (**B**) Vulval morphology in wild type (left) versus a *qy1* mutant (right). The number (see text) and positions of vulF, vulE, and vulD (outlined in CAAX::GFP) were normal in *lin-29(qy1)* animals (top panels: grayscale, bottom panels: green), while the BM (laminin::mCherry, middle panels: grayscale and bottom panels: magenta) gap boundaries are narrower in the *lin-29(qy1)* mutant. (bracket). The arrow in the middle panel for the wild type denotes a puncta of laminin internalized by the AC during invasion, and does not represent the edge of the BM gap (which is located above vulD in wild type and is characterized by a build-up of laminin). All images lateral, central plane. Scale Bars, 5 um.

*Figure 2 continued on next page*

*Figure 2 continued*

The following figure supplement is available for figure 2:

**Figure supplement 1.** Vulval morphology measurements in lin-29 mutants.

height and width (*Figure 2—figure supplement 1*)—was the same in *lin-29(qy1)* mutants compared with wild type animals. Taken together, these results indicate that *lin-29* mutants have a specific defect in BM sliding that is independent of the role of the vulval cell divisions in controlling BM movement (*Matus et al., 2014*).

## The LIN-29 protein is expressed and functions in the AC to promote BM sliding

To determine where LIN-29 functions to promote BM sliding, we first examined the expression pattern of the *lin-29* gene. We fused 5-kb of the 5' cis-regulatory element of the *lin-29* gene to GFP (*lin-29a/b > GFP*; see Experimental Procedures). Using this reporter, we detected expression of *lin-29* in the AC throughout the course of uterine-vulval connection (*Figure 3A*), but did not detect expression in the other cells in contact with or bordering the BM boundary. This expression pattern is consistent with published immunofluorescence studies showing that the LIN-29 protein is expressed at high levels in the AC during this time (*Bettinger et al., 1997*). We also fused GFP to the 5-kb region upstream of the *lin-29c* isoform, but did not see any expression in the uterine or vulval cells at the time of uterine-vulval connection. These results suggest that the *lin-29a/b* isoforms might act in the AC to promote BM sliding.

To further test the idea that that the *lin-29a/b* isoforms have a specific function in BM sliding, we scored *lin-29(ga94)* mutants. The *ga94* mutation creates an early stop codon in exon IV of the *lin-29a* and *lin-29b* isoforms but removes only the first two amino acids of the shorter *lin-29c* isoform (see *Figure 1B*). Previous studies have reported that *lin-29(ga94)* mutants display some, but not all of the phenotypes associated with *lin-29* null alleles (*Bettinger et al., 1997*). We measured BM gap expansion and found that *ga94* mutants had a similar defect in the BM gap opening to *qy1* mutants (*Figure 3B*). These results suggest that LIN-29A and/or LIN-29B—but not LIN-29C—promote BM gap expansion. We next performed an AC-specific rescue of *lin-29a* expression in *lin-29(qy1)* mutant animals using an AC-specific promoter (*cdh-3 > lin-29a::GFP*; [*Kirouac and Sternberg, 2003*]). We found that LIN-29A::GFP protein expressed only in the AC completely rescued BM gap expansion (*Figure 3C,D*). We conclude that LIN-29 functions within the AC to promote BM sliding.

## LIN-29 does not promote AC de-adherence from the BM to allow BM sliding

As the BM remained in contact with the AC in most *lin-29* mutants, we hypothesized that LIN-29 might be required for AC de-adhesion from the BM. We used laser-directed ablation to specifically kill the AC in *qy1* mutants immediately after AC invasion and then assessed BM gap expansion. We found that the BM gap still failed to open in the absence of the AC in *lin-29* mutant animals (n = 8/8 animals, *Figure 4A*). Ablation of the AC at this stage in wild type animals does not prevent BM sliding and gap expansion (*Ihara et al., 2011*). These results suggest that LIN-29 does not facilitate sliding by causing the AC to de-adhere from the BM.

## LIN-29/AC-mediated uterine π cell fate specification is required for BM sliding

We next hypothesized that the role of LIN-29 in the AC might be to regulate BM sliding through interactions with the neighboring uterine cells at the BM gap boundary. During the early L3 stage (just prior to the time of AC invasion), LIN-29 is required in the AC to specify the π cell fate in the six ventral uterine cell descendants that directly neighbor the AC (*Newman et al., 2000*; *Newman, 1995*). LIN-29 maintains the expression of the transmembrane Notch ligand LAG-2 in the AC, which activates LIN-12 in the neighboring ventral uterine cells and specifies the π cell fate (*Newman et al., 2000*; *Newman, 1995*). We thus investigated whether the π cell fate might be required for BM sliding.

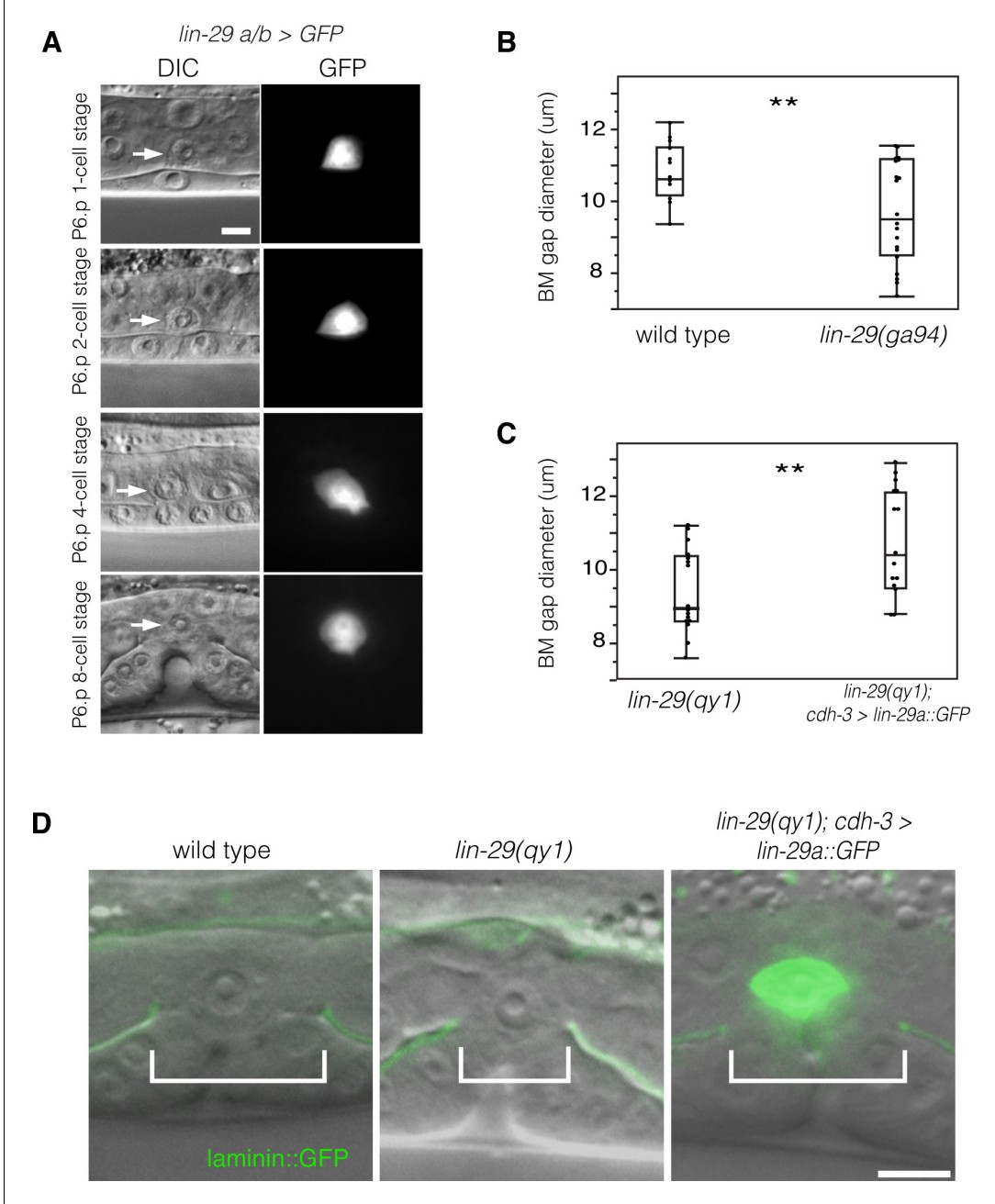

**Figure 3.** LIN-29 functions within the AC to promote BM gap expansion. (**A**) Expression of GFP from a *lin-29a/b > GFP* transcriptional reporter in the AC throughout the course of BM gap expansion. (**B**) Quantification of the BM gap in wild type (n = 16) and *lin-29(ga94)* (n = 22) at the P6.p 8-cell stage. (**C**) Quantification of the BM gap in *lin-29(qy1)* (n = 16) and AC specific rescue of LIN-29A (n = 15) at the P6.p 8-cell stage. (**D**) Laminin::GFP overlaid on DIC in wild type, *lin-29(qy1)*, and a *lin-29* mutant with AC specific rescue of LIN-29A protein showing diameter of the BM gap (brackets). In B and C, ** indicates p<0.01, Wilcoxon rank sum test. All images lateral, central plane. Scale Bars, 5 um.

The following source data is available for figure 3:

**Source data 1.** BM gap diameter in wild type vs. *lin-29(ga94)*.

**Source data 2.** BM gap diameter in *lin-29(qy1)* vs. *lin-29(qy1); cdh-3 > lin-29a::GFP*.

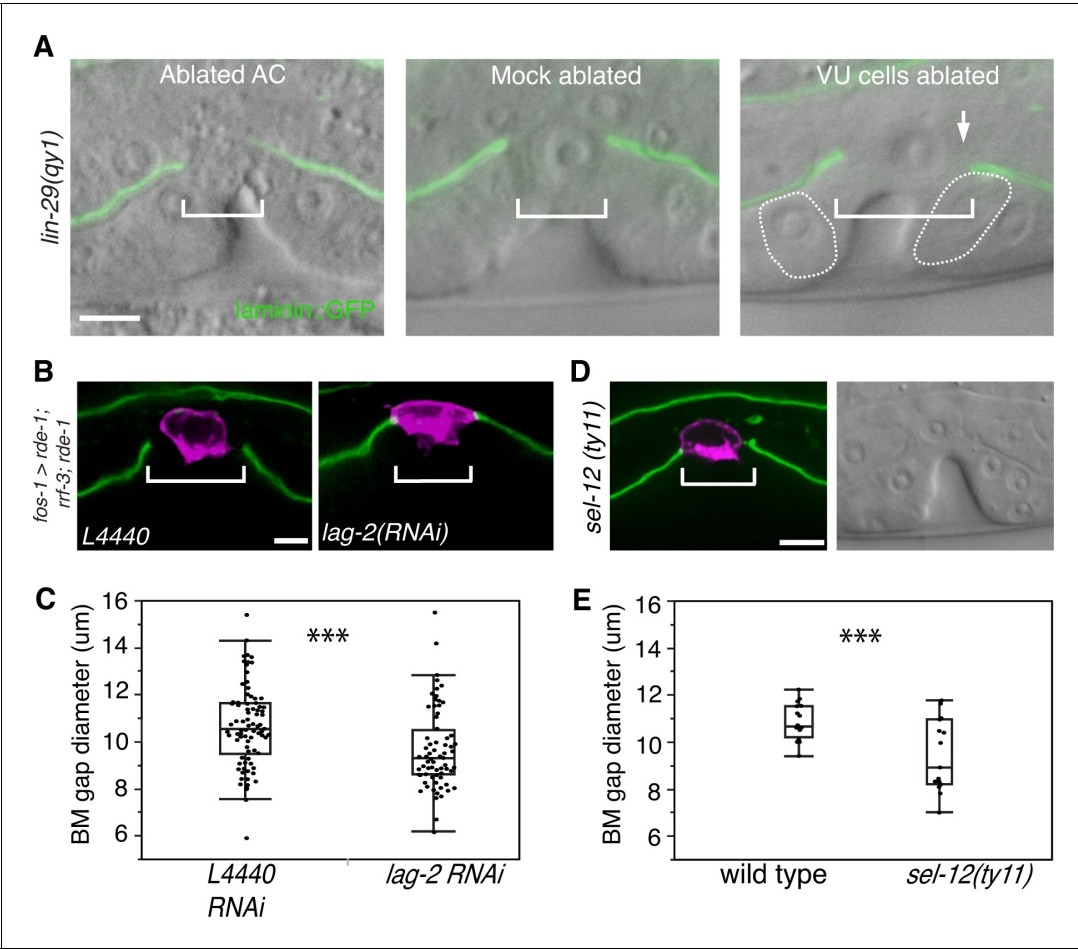

**Figure 4.** AC-mediated Notch signaling to the uterine π cells promotes BM sliding. (**A**) Fluorescence overlays of laminin::GFP on DIC images show that the BM gap (brackets) failed to expand in *lin-29(qy1)* mutants after the AC was ablated (left, n = 8/8) or in mock ablated *lin-29* mutants (middle, n = 8/8). However, the BM gap opened wider on the side of the AC in *lin-29* mutants (BM slid over the vulD cell, outlined with dashed line) after laser ablation of the uterine π cells on that side (right, arrow, n = 5/8 animals; p<0.001, Fisher's Exact Test). (**B**) Uterine-specific RNAi targeting the L4440 empty vector control (left) and the AC-expressed Notch ligand *lag-2* (right) revealed that *lag-2* is required for BM sliding (BM visualized with laminin::GFP, green) away from the AC (*mCherry::PLCδ^{PH}*, magenta). (**C**) Quantification of the BM gap size after treatment L4440 control (n = 62) and uterine-specific *lag-2* RNAi (right, n = 68). (**D**) In *sel-12(ty11)* mutants, which perturbs LIN-12(Notch) signaling in the uterine π cells, the BM gap (laminin::GFP, right; visualized at P6.p 8-cell stage, DIC left) failed to expand. (**E**) Quantification of the BM gap in *sel-12(ty11)* (n = 15) and wild type (n = 15). For C and E, *** indicates p<0.001 Wilcoxon rank sum test. All images lateral, central plane. Scale Bars, 5 μm

The following source data and figure supplement are available for figure 4:

**Source data 1.** BM gap diameter in *L4440* vs. *lag-2* RNAi.
**Source data 2.** BM gap diameter in wild type vs. *sel-12(ty11)*.
**Source data 3.** BM gap diameter in *lag-2* vs. *lin-12 and glp-1* RNAi.
**Figure supplement 1.** The AC signals through Notch receptor LIN-12 to promote BM sliding.

We perturbed π cell fate specification by knocking down *lag-2* expression specifically in the uterine cells. Uterine-specific RNAi was attained with uterine-specific expression of the Argonaute/PIWI gene *rde-1* in an *rde-1* mutant background, which restores RNAi only in the uterine tissue (*Hagedorn et al., 2009*). Uterine *lag-2* knockdown recapitulated the BM sliding defect seen in *lin-29* mutants (*Figure 4B,C*), suggesting that the LIN-29 protein acts in a LAG-2 dependent manner to

promote BM sliding. Notably, the reduction in BM gap opening after *lag-2* RNAi was less than in *lin-29* mutants (*Figure 4C* versus *Figure 3C*), which was likely due to the incomplete loss of *lag-2* by RNAi. Consistent with LAG-2 acting through the LIN-12 (Notch) receptor, we also used RNAi targeting *lin-12* and *glp-1* (the only other encoded Notch receptor in the *C. elegans* genome) (*Greenwald, 2005*) and found that loss of *lin-12*, but not *glp-1*, phenocopied *lag-2* RNAi (*Figure 4—figure supplement 1*). Finally, we assessed gap expansion in mutants that lacked the ability to transduce Notch signaling in the π cells (*sel-12(ty11)*). SEL-12 is a *C. elegans* presenilin that cleaves ligand bound LIN-12 (Notch) so that the intracellular domain is released to translocate to the nucleus and activate target gene expression. No other *C. elegans* presenilins act in the prospective π cells (*Cinar et al., 2001*), so loss of *sel-12* specifically targets π cell specification and circumvents other phenotypes associated with the absence of Notch signaling. Loss of *sel-12* led to a similar defect in BM gap expansion as *lin-29* mutants (*Figure 4D,E*). These observations offer strong evidence that LIN-29 in the AC regulates BM sliding by specifying ventral uterine descendents adjacent to the BM gap boundary to adopt the π cell fate.

## Ablation of the ventral uterine cells restores BM sliding in *lin-29* mutants

We next hypothesized that LIN-29 mediated specification of the π cells was required for them to reduce their adhesion from the BM. If this were the case, removal of the unspecified uterine boundary cells should restore BM sliding in *lin-29* mutants. To test this notion, we ablated the precursors of the π cells (ventral uterine cells) on one side of the AC in *lin-29(qy1)* mutants. We found that this treatment rescued the BM sliding defect on that side (n = 5/8 animals, *Figure 4A*). These results support the idea that π cell fate specification leads to reduced adhesion to the BM, which facilitates BM sliding.

## Uterine π cell divisions are not required to facilitate BM sliding

When the vulval cells vulE and vulF divide at the BM gap boundary, the cells round and briefly reduce their attachment to the BM. This loss of attachment during division promotes BM movement over vulE and vulF, which then stops at the non-dividing vulD cell (*Matus et al., 2014*). We thus hypothesized that the uterine π cell divisions, which occur near the time of primary vulval cell divisions (see *Figure 1A* and *Figure 5A*), might be coordinated with the vulE and vulF divisions to facilitate the sliding of the linked ventral and gonadal BMs.

During the time of BM gap expansion (P6.p 4-cell to 6-cell transition) each π cell divides once along the dorsal-ventral axis (*Figure 5A*). To examine the precise timing of these π cell divisions in relation to the vulval divisions, we used a membrane bound GFP that marks both the VPCs and the uterine cells (CED-10::GFP; [*Ziel et al., 2009*]). While the π cells began dividing near the same time as the vuE and vulF cells, we found little correlation between the timing of their divisions (n = 8/46 π cells in 13 animals dividing coincidently with vulE on the other side of the BM, n = 16/54 π cells in 12 animals dividing coincidently with vulF on the other side of the BM, *Figure 5B*). These observations suggest that coordination of π cell divisions with vulE and vulF divisions are not required for BM sliding.

To directly determine if π cell divisions are required to expand the BM gap, we blocked their division by expressing the cyclin dependent kinase inhibitor *cki-1* driven with an early π cell specific promoter (*egl-13 > cki-1::GFP*) in wild type animals (*Figure 5C*). *egl-13* initiates expression at the time of π cell birth when there are six π cells in the late L3 stage (*Cinar et al., 2003*). We found no difference in the BM gap size at the P6.p 8-cell stage between animals (*Hanna-Rose and Han, 1999*) in which π cell divisions had been arrested compared to wild type controls (*Figure 5D*). We conclude that π cell divisions are not required for these BM boundary cells to loosen their adhesion to the BM to promote BM sliding.

## π cells do not completely lose contact with the BM during BM sliding

We next examined the relationship of the BM to the π cells during BM sliding and stabilization. Notably, on the vulval side of the developing uterine-vulval connection, the BM moves completely over the dividing vulE and vulF cells and the edge of the BM gap stabilizes on the non-dividing vulD cell (*Matus et al., 2014*). In contrast, we found that as the BM slid along the π cells to expand the

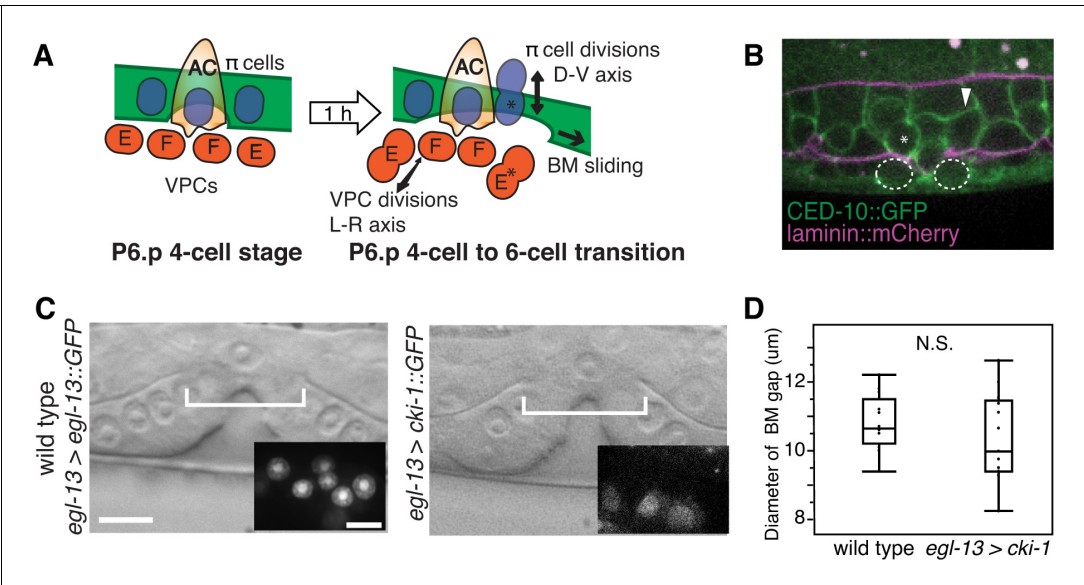

**Figure 5.** π cell division does not regulate BM sliding. (A) A schematic diagram presenting the hypothesis that uterine π cell (blue) divisions (along the dorsal-ventral axis) are coordinated with divisions of the underlying vulE and vulF cells (red, which divide along the left-right axis) such that the BM (green) is released on both sides when dividing cells lose contact with the BM, allowing it to slide. (B) Fluorescence overlay showing a cell membrane marker (CED-10::GFP, green) of dividing vulE (dashed outline) and adjacent uterine π cells (asterisk denotes a dividing cell and arrowhead a non-dividing cell) with the BM (laminin::mCherry, magenta). (C) A lateral, central plane DIC image of wild type (left) with inset showing divided π cells expressing EGL-13::GFP driven by the *egl-13* promoter (*egl-13 > egl-13::GFP*) had a similar BM gap size (bracket) to an animal expressing CKI-1::GFP in the π cells (right, *egl-13 > cki-1::GFP*), thus arresting their divisions. (D) Quantification of the BM gap in wild type (n = 15) and *egl-13 > cki-1::GFP* transgenic animals (n = 13). Only animals in which the π cells failed to divide were included in the analysis. No significant difference was observed (p>0.05, Wilcoxon rank sum test).

The following source data is available for figure 5:

**Source data 1.** BM gap diameter in wild type vs. *egl-13 > cki-1*.

gap, the edge of the BM gap stabilized on the π cells; BM maintained contact with a significant portion the ventral π cells (n = 13/13 animals imaged during the BM gap expansion, *Figure 6A–C*). These observations indicate that, in contrast to the vulval cells that lose attachment to the BM, the uterine π cells remain in contact with and are possibly adherent to the BM as the BM slides.

## Notch promotes BM sliding by upregulating *ctg-1* (Sec14-GOLD protein) in the π cells

Based on our observations, we hypothesized that LIN-29-mediated LIN-12/Notch activation in the π cells may result in transcriptional changes in the π cells that reduce but do not eliminate cell-BM adhesion. During LIN-12/Notch activation, the intracellular domain of LIN-12/Notch is cleaved and enters the nucleus where it forms a complex with the DNA binding protein LAG-1 (the worm ortholog of CSL) that promotes expression of LIN-12/Notch target genes (*Greenwald, 2012*). Bioinformatic studies have previously identified 163 putative direct targets of LIN-12/Notch that contain clusters of LAG-1 binding sites (*Yoo, 2004*). We conducted an RNAi screen of 104 of the 163 presumptive Notch targets to assess BM gap opening. Worms were scored as having a small BM gap diameter if the BM remained in contact with the AC or vulF cells at the P6.p 8-cell stage—a time when the BM stabilizes over vulD in control animals. Through this screen, we found that RNAi-mediated knock down of *ctg-1*, which encodes a Sec14-GOLD protein, a poorly understood group of the Sec14 family of phospholipid transfer proteins (*Mousley et al., 2007*; *Saito et al., 2007*), led to a similar BM sliding defect as knock down of the LIN-12/Notch ligand *lag-2* (*Supplementary file 1*). Furthermore, we found that RNAi-mediated knock down of *ctg-1* expression in a uterine-specific RNAi strain also resulted in a BM sliding defect and smaller BM opening

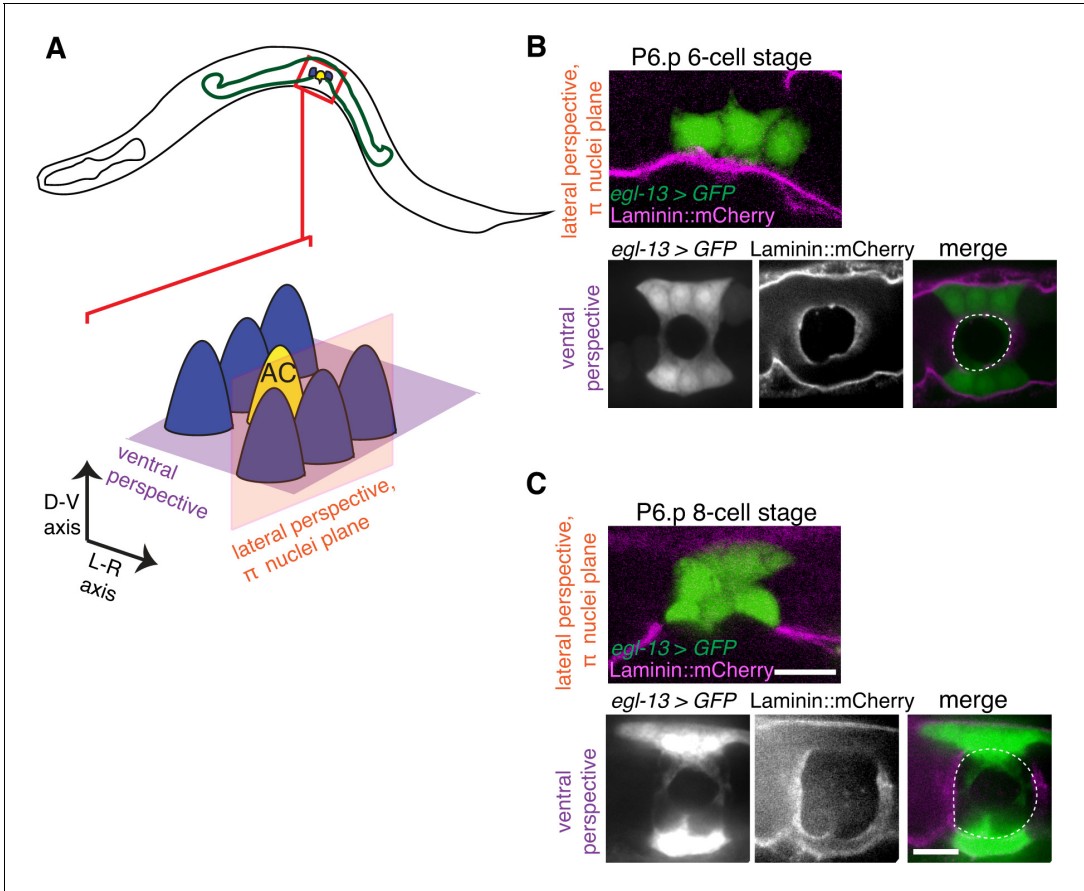

**Figure 6.** The π cells remain in contact with the BM during and after BM sliding. (A) A schematic diagram showing image planes of lateral view (orange) and ventral view (purple) of the π cells in relation to the BM. (B) In the early P6.p 6-cell stage, the π cells (visualized with GFP driven by the *egl-13* promoter, green) lie almost completely over BM (magenta). (C) By the P6.p 8-cell stage, the BM slides open farther and the π cells loose BM contact in the central region, but maintain contact on the lateral edges of the cell. Scale bars, 5 μm. In B and C, ventral perspective is a max intensity projection of all z-slices with the BM gap edge and the edge of the BM highlighted in merge (white dashed line).

(*Figure 7A*). These results are consistent with the possibility that *ctg-1* is a primary target of LIN-12/ Notch signaling in the π cells and mediates BM sliding.

The upstream regulatory region of *ctg-1* has 19 LAG-1 binding sites (*Figure 7—figure supplement 1*), which is why it is predicted to have direct transcriptional regulation by LIN-12/Notch signaling. To determine if *ctg-1* is upregulated in the π cells, we created a transcriptional reporter for *ctg-1* by replacing the majority of the coding region with GFP using CRISPR/Cas9 genome editing (*Figure 7—figure supplement 1*; *ctg-1 > GFP*; *qy11* allele) (*Dickinson et al., 2013*). Strikingly, we found *ctg-1 > GFP* was specifically upregulated in the π cells during the time of BM sliding in wild type animals (*Figure 7B*), but not *lin-29* mutants (*Figure 7B,C*). Furthermore, as this GFP insertion created a *ctg-1* loss of function mutation, we examined BM sliding in *qy11* worms and confirmed that genetic loss of *ctg-1* led to a reduction in BM sliding and gap opening (*Figure 7D,E*). To assess whether restoration of CTG-1 in the ventral uterine cells in a *lin-29* mutant could restore BM sliding, we expressed a full-length GFP translational fusion of *ctg-1* under the control of a pan-uterine promoter (*fos-1 > ctg-1::GFP*) in *lin-29* mutant animals. CTG-1::GFP predominantly localized to the cytosol of the uterine cells (*Figure 7—figure supplement 2*), consistent with reports of Sec14 family protein localization in yeast, plant and mammalian cells (*Schnabl et al., 2003*; *Shibata et al., 2001*). Restoration of uterine expression of CTG-1::GFP in *lin-29* mutants rescued BM sliding and gap opening (*Figure 7F,G*). Taken together, these results offer compelling evidence that *ctg-1* is a direct and

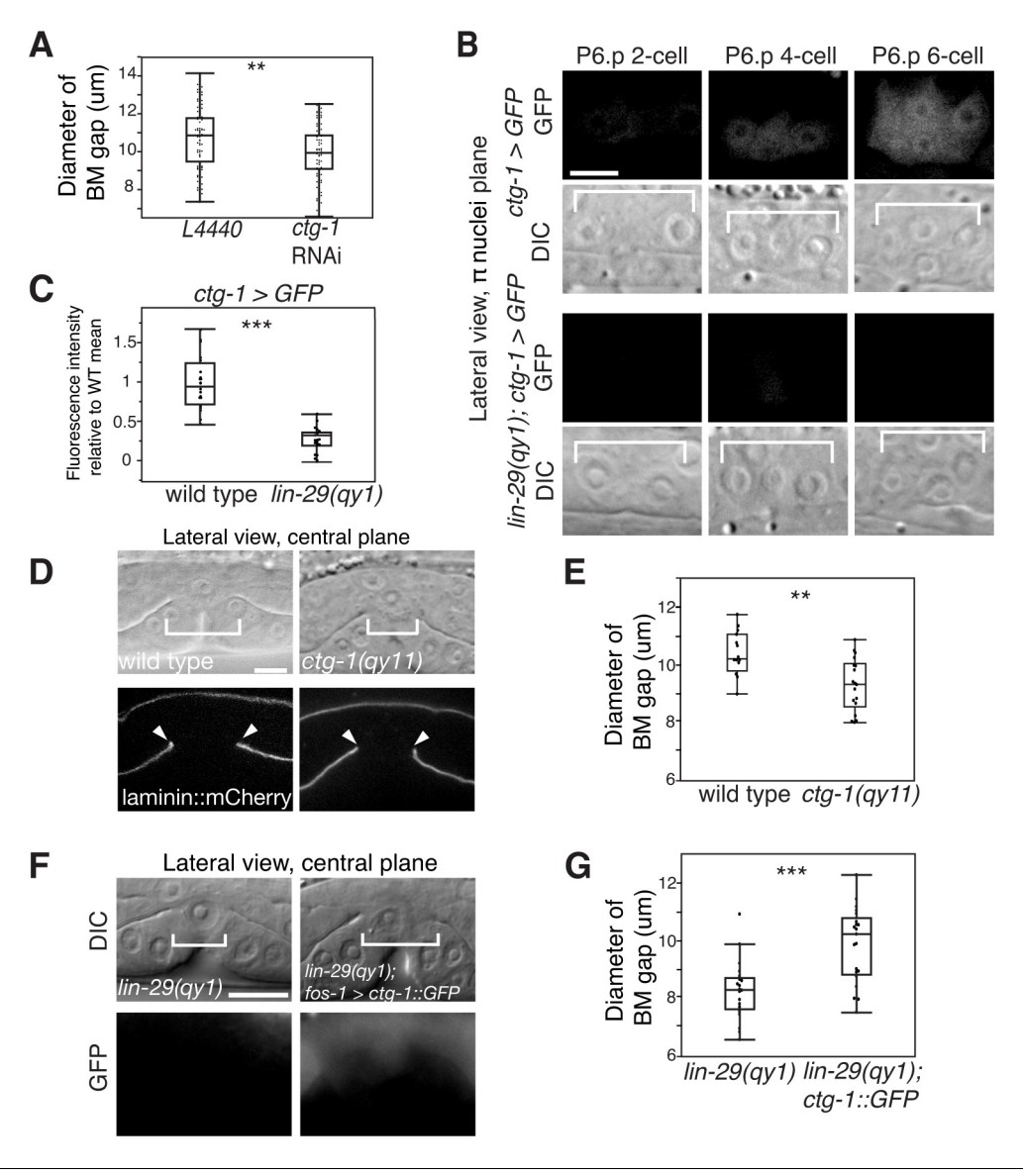

**Figure 7.** Notch signaling promotes BM sliding through upregulation of the Sec14 family phospholipid transfer protein CTG-1. (**A**) Quantification of uterine-specific RNAi targeting *ctg-1*. Loss of *ctg-1* led to a smaller BM gap at the early L4 (L4440 control, n = 65; *ctg-1* RNAi, n = 64). (**B**) DIC and fluorescence images showing lateral views in the π cell nuclei plane (see *Figure 5F*) of animals harboring a genomic insertion of GFP at the *ctg-1* locus (*ctg-1 > GFP*). *ctg-1* expression is upregulated in the π cells of wild type worms at the time of Notch mediated induction of π cell fate, which coincides with BM sliding (n = >5 animals each stage, top). Expression of *ctg-1* was not induced when the π cell fate was disrupted in a *lin-29* mutant (n = >5 animals each stage; bottom). Brackets in DIC images denote the location of the π cells in wild type and non-specified ventral uterine boundary cells in *lin-29(qy1)* animals. (**C**) Quantification of fluorescence intensity of the GFP from the *ctg-1> GFP* genomic locus shows reduced expression in *lin-29* mutants relative to wild type (n = 20 for each). Values in the plot are normalized to the wild type mean. The asterisks signify a statistically significant difference. (**D**) A CRISPR-cas9 mediated deletion allele of *ctg-1(qy11)* (lateral central plane; DIC, top; laminin::mCherry, bottom) has a smaller BM gap (bracket and arrowheads) compared to a wild type animal. (**E**) Quantification of BM gap diameter in *ctg-1(qy11)* (n = 18) and wild type (n = 16) animals. (**F**) Pan-uterine cell expression of CTG-1 (*fos-1 > ctg-1::GFP*) in a *lin-29(qy1)* mutant increased BM gap size (brackets). Lateral central plane; DIC, Top; GFP expression, bottom. (**G**) Quantification of BM gap diameter in *lin29(qy1)* (n = 21) and *lin-29(qy1)* rescued with *fos-1 > ctg-1::GFP* (n = 21). For **A**, **C**, **E**, and **G**, the asterisks signify a statistically significant difference (** indicates p<0.01 and *** indicates p<0.001, Wilcoxon rank sum test). Scale bars, 5 μm.

*Figure 7 continued on next page*

*Figure 7 continued*

The following source data and figure supplements are available for figure 7:

**Source data 1.** BM gap diameter in *L4440* vs. *ctg-1* RNAi.
**Source data 2.** Fluorescence intensity of GFP from *ctg-1 > GFP* in wild type vs. *lin-29(qy1)*.
**Source data 3.** BM gap diameter in wild type vs. *ctg-1(qy11)*.
**Source data 4.** BM gap diameter in *lin-29(qy1)* vs. *lin-29(qy1); fos-1 > ctg-1::GFP*.
**Source data 5.** BM gap diameter in L4440 vs. *pifk-1*, C56A3.8, ZC8.6, Y75B8A.24, and *ppk-1* RNAi.
**Figure supplement 1.** LAG-1 binding sites in the *ctg-1* promoter and *ctg-1(qy11)* schematics.
**Figure supplement 2.** CTG-1::GFP expression.
**Figure supplement 3.** PI4-kinase PIFK-1 is required for BM sliding.

crucial transcriptional target of LIN-29-mediated LIN-12/Notch signaling in the π cells that promotes BM sliding.

## LIN-29/CTG-1 reduces dystroglycan localization and trafficking in the π cells

Sec14 domain-containing genes are a eukaryotic-specific gene family. Six Sec14 family proteins are encoded in the *S. cerevisae* genome and more than 20 exist in vertebrates (*Bankaitis et al., 2010*). We found that at least 16 Sec14 family members are encoded in the *C. elegans* genome (*Supplementary file 2*), with 12 encoding Sec14-GOLD proteins. Evidence suggests that Sec14 family proteins act as site-specific catalysts of phosphatidylinositol 4-OH kinases that generate phosphatidylinositol-4-phosphate (PI(4)P) (*Bankaitis et al., 2010*; *Huang et al., 2016*; *Schaaf et al., 2008*). Many Sec14 family proteins are thought to regulate vesicular trafficking, yet very little is known about the precise cellular processes controlled by most of these proteins (*Curwin et al., 2009*; *Grabon et al., 2015*; *LeBlanc and McMaster, 2010*). In *S. cerevisiae* Sec14 regulates trafficking through the PI4-kinase Pik1. To determine whether CTG-1 might regulate BM sliding through a PI4-kinase, we examined uterine-cell specific RNAi-mediated knockdown of the four PI4-kinases encoded in the *C. elegans* genome. We also examined knockdown of the sole *C. elegans* PI5-kinase, *ppk-1*, which uses PI(4)P to generate PI(4,5)P$_2$ (*Weinkove et al., 2008*). In support of a similar role for CTG-1 as Sec14, we found that reduction of *pifk-1*(ortholog of Pik1 and vertebrate PI4KIIIβ), but not other PI4-kinases or *ppk-1*, decreased BM sliding and gap expansion (*Figure 7—figure supplement 3*). As Sec14 and Pik1/ PI4KIIIβ are implicated as regulators of vesicle trafficking (*Clayton et al., 2013*; *Sciorra et al., 2005*), we hypothesized that *ctg-1* might be upregulated in the π cells to alter the trafficking and localization of BM adhesion receptor proteins to promote BM sliding.

One of the major BM adhesion receptors in *C. elegans* is integrin. All integrins are heterodimers composed of a single α and β subunit. *C. elegans* have two heterodimers, composed of αPAT-2 or αINA-1 subunit bound to the sole β subunit, PAT-3 (*Kramer, 2005*). Only the INA-1/PAT-3 heterodimer is expressed in uterine and vulval tissue at the time of uterine-vulval attachment. INA-1/PAT-3 integrin is necessary for the stabilization of the BM over vulD and its loss results in the Pvl phenotype (*Hagedorn et al., 2009*; *Ihara et al., 2011*). We did not, however, find any changes in the expression or localization of the functional integrin reporter PAT-3::GFP (*Hagedorn et al., 2009*) in the *lin-29(qy1)* mutant background, which lacks *ctg-1* expression, compared to wild type (*Figure 8—figure supplement 1*). Additionally, fluorescence recovery after photobleaching (FRAP) experiments indicated that PAT-3::GFP was trafficked to the cell-BM interface normally at the time of BM sliding (between P6.p 4-cell and 8-cell stages) in *lin-29* mutants (*Figure 8—figure supplement 1*). Finally, RNAi-mediated targeting of *ina-1* in the uterine-specific RNAi background did not rescue BM sliding

in *lin-29* mutants (BM gap diameter 8.13 ± 0.16 μm (L4440 empty vector control, n = 48), 8.30 ± 0.25 μm [*ina-1* RNAi, n = 48]). These experiments strongly suggest that CTG-1 does not regulate INA-1/PAT-3 localization in the π cells to promote BM sliding.

We next examined the expression and localization of DGN-1, the sole *C. elegans* ortholog of the BM receptor dystroglycan. DGN-1 is expressed in the uterine tissue and, like integrin, its loss results in a Pvl phenotype, indicating potential functions in uterine-vulval attachment (*Johnson et al., 2006*). We detected a significant increase in the localization of DGN-1 at the cell-BM interface in the π cells of *lin-29* mutants compared to wild type animals (*Figure 8A,B*). This difference was not a result of increased expression, as the transcriptional reporter for *dgn-1* was similarly expressed in *lin-29* mutants and wild type (*Figure 8C*). In addition, the localization of DGN-1 in regions beyond the edges of the BM gap (over the vulD cell) were similar in *lin-29* and wild type, indicating that the alteration in DGN-1 localization was specific to the region of BM sliding near the AC (*Figure 8—figure supplement 2*). FRAP experiments indicated that DGN-1 recovered more quickly after photobleaching in *lin-29* mutants versus wild type animals, which may account for the increased DGN-1:: GFP localization (*Figure 8F,G*). Faster recovery of DGN-1 at the cell-BM interface in *lin-29* mutants could indicate more rapid cell surface trafficking, slower endocytosis/removal, or faster lateral plasma membrane mobility. Notably, the number of DGN-1 containing Rab-7 marked late endosome vesicles in *lin-29* mutants and wild type animals was similar, suggesting that CTG-1 might not direct rapid removal of DGN-1 from the cell surface into the endosomal system (*Figure 8—figure supplement 3*). There was, however, a reduction of F-actin at the basal surface in the π cells in *lin-29* mutants (*Figure 8—figure supplement 4*). As F-actin is thought to act as a barrier to exocytosis (*Eitzen, 2003*), these observations support the idea of increased DGN-1 delivery to the cell surface. Given the association of the GOLD domain, Sec14 proteins, and PI4-kinase regulation with vesicle trafficking, (*Anantharaman and Aravind, 2002*; *Clayton et al., 2013*; *Mousley et al., 2007*), CTG-1 may direct a specific trafficking pathway that restricts the delivery of DGN-1 to the cell-BM interface. Importantly, these observations do not rule out the possibility that CTG-1 enhances endocytic removal from the cell surface at levels that we were unable to detect or that it regulates the membrane mobility of DGN-1 to limit DGN-1 at the cell-BM interface.

We next determined if the increase of DGN-1 at the cell-BM interface in the *lin-29* mutant could be responsible for stronger adhesion of the uterine cells to the BM, causing BM sliding to fail. Consistent with this idea, reduction of *dgn-1* expression via uterine specific RNAi in the *lin-29(qy1)* mutant restored BM sliding and gap expansion (*Figure 8D,E*). Taken together, these results suggest that LIN-29 activity in the AC leads to LAG-2/LIN-12 (Notch)-mediated upregulation of *ctg-1* expression in the uterine π cells that restricts the cell surface trafficking and accumulation of the BM receptor DGN-1, which facilitates BM sliding during uterine-vulval attachment.

## Discussion

The shifting of cell-BM interfaces is crucial in many diverse morphogenetic events, including intestinal epithelial renewal, branching morphogenesis, and BM deposition (*Clevers, 2013*; *Glentis et al., 2014*; *Morrissey and Sherwood, 2015*). How BM sliding is controlled, however, remains poorly understood. We show here that the AC in *C. elegans* further widens a breach it creates in BM during uterine-vulval attachment by promoting BM sliding in neighboring uterine cells. LIN-29 maintains the expression of the Notch ligand LAG-2 in the AC, which leads to the activation of Notch signaling and induction of uterine π cell fate in the neighboring cells that sit over the nascent BM gap boundary. Notch activation in turn leads to upregulation of the Sec14 family phospholipid transfer protein CTG-1 in these BM gap boundary cells, which restricts trafficking of the receptor DGN-1 (dystroglycan) to the cell-BM interace. Our data suggest that the proper regulation of DGN-1 trafficking modulates cell-BM adhesion and allows the BM slide, which further widens the BM breach. This work reveals a new morphogenetic signaling pathway that promotes cell-BM sliding (See Summary *Figure 9*).

During cell-BM sliding, the cells, the BM, or both shift position in relationship to each other. During uterine-vulval attachment in *C. elegans*, the linked ventral and gonadal BMs slide over vulval and uterine cells to expand the BM gap, and the vulval cells also appear to move (through migration and invagination) inside the gap to form direct contacts with uterine cells (*Ihara et al., 2011*; *Schindler and Sherwood, 2013*). It has previously been shown that vulval cell divisions at the BM

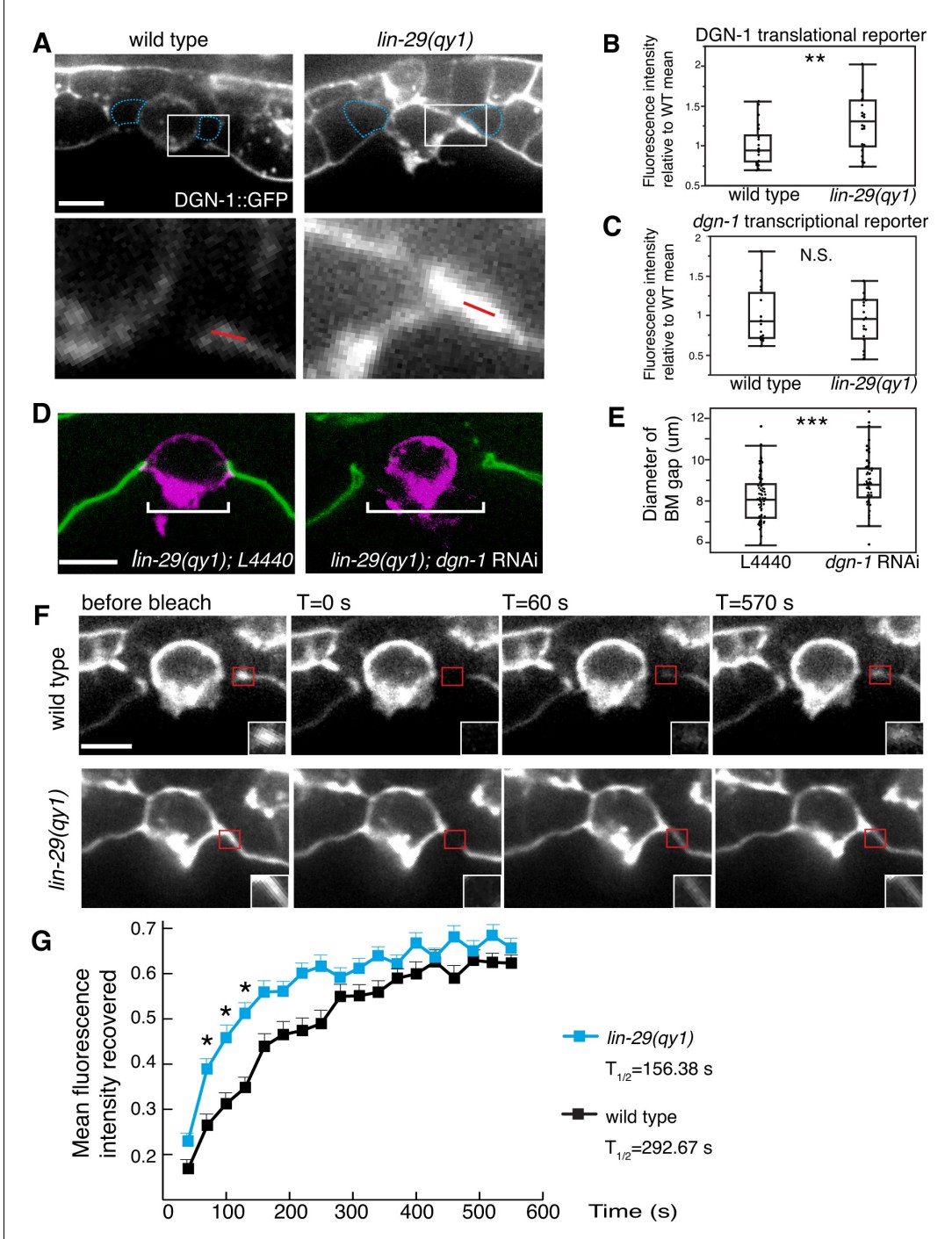

**Figure 8.** LIN-29/CTG-1 restricts dystroglycan (DGN-1) levels and trafficking at the π cell-BM interface. (A) A functional translational fusion of the BM-receptor DGN-1 (DGN-1::GFP, top) reveals that more DGN-1 localizes on the surface of the ventral uterine cells (blue dashed outlines) at the edges of the BM gap in a *lin-29* mutant compared to a wild type animal. Enlargement of boxes in top images (bottom) shows regions (red bars) where DGN-1 levels were measured in (B). (B) Quantification of the fluorescence intensity of a functional DGN-1::GFP translational reporter (*dgn-1 > dgn-1::GFP*) at the BM gap boundary in the π cells in wild type and *lin-29(qy1)* animals (n = 22 for each). Values in plot are normalized to the wild type mean (** indicates p<0.01, Wilcoxon rank sum test). (C) Quantification of the fluorescence intensity of a *dgn-1* transcriptional reporter in wild type and *lin-29(qy1)* animals (n = 21 for each) revealed no significant difference in expression levels (p>0.05 Wilcoxon rank sum test). Values in plot are normalized to the wild type mean. (D) Uterine-specific *dgn-1* RNAi rescued the BM gap (brackets) defect in a *lin-29(qy1)* background expressing BM and AC reporters (laminin::GFP and *cdh-3 > mCherry::PLCδ^{PH}*). (E) BM gap diameters in uterine-specific RNAi knockdown of *dgn-1* (n = 50) and L4440 control (n = 63) in the *lin-29* mutant background (*** indicates p<0.001, Wilcoxon rank sum test). (F) Fluorescence recovery after photobleaching (FRAP) of a 1.0-μm

*Figure 8 continued on next page*

*Figure 8 continued*

region of DGN-1::GFP along the basal surface of the uterine π cells during the late the early P6.p 6-cell stage (time of BM sliding) in wild type and *lin-29 (qy1)* mutant animals. DGN-1::GFP recovers more quickly in the *lin-29* mutant. (G) Graph reports DGN-1::GFP recovery in wild type and *lin-29(qy1)* mutants (n = 9 animals for each). Error bars represent standard deviation and the asterisks signify a statistically significant difference at comparable time points (p<0.05 Student's *t*-test). $T_{1/2}$ denotes time to recovery of 50% of initial fluorescence signal. All images lateral, central plane. Scale bars, 5 µm.

The following source data and figure supplements are available for figure 8:

**Source data 1.** Fluorescence intensity of DGN-1::GFP in wild type vs. *lin-29(qy1).*

**Source data 2.** Fluorescence intensity of GFP from a *dgn-1 > GFP* transcriptional reporter in wild type vs. *lin-29(qy1)* animals.

**Source data 3.** BM gap diameter in *L4440* vs. *dgn-1* RNAi in *lin-29(qy1)* background.

**Source data 4.** Fluorescence intensity recovered over time in DGN-1::GFP FRAP.

**Source data 5.** Fluorescence quantifications for *pat-3* transcriptional and translational markers.

**Source data 6.** Fluorescence intensity recovered over time in PAT-3::GFP FRAP.

**Source data 7.** Fluorescence intensity of DGN-1::GFP in wild type vs. *lin-29(qy1)* over vulD (past region of BM sliding).

**Source data 8.** RAB-7 and DGN-1 co-localization in wild type vs. *lin-29(qy1).*

**Source data 9.** Fluorescence intensity of moesinABD::mCherry in wild type vs. *lin-29(qy1).*

**Figure supplement 1.** Integrin expression and localization are unaffected by LIN-29/CTG-1.

**Figure supplement 2.** Loss of LIN-29/CTG-1 does not affect dystroglycan (DGN-1) localization in regions beyond the BM gap interface.

**Figure supplement 3.** Loss of LIN-29/CTG-1 does not affect DGN-1 localization to late endosomes.

**Figure supplement 4.** F-actin is reduced at the cell-BM interface with loss of LIN-29/CTG-1.

gap boundary controls BM sliding (*Matus et al., 2014*). During vulval invagination and growth, the centrally located vulE and F cells divide and reduce their contact with the BM during cell rounding, allowing the BM to slide over these cells. The BM ultimately ceases sliding and the BM gap boundary stabilizes on the non-dividing vulD cell. Blocking the divisions of vulE and F halts BM sliding prematurely resulting in a narrower BM gap, while inducing vulD divisions causes the BM to slide further and enlarges the opening (*Matus et al., 2014*). Here we have found that uterine cells at the BM gap boundary regulate BM movement through a distinct mechanism independent of cell division. Uterine π cell divisions are not coordinated with vulval cell divisions during BM movement. Further, blocking uterine π cell divisions did not halt BM sliding. Instead, through a forward genetic screen, site of action, and expression studies, our data indicate that the invading AC directs a morphogenetic signaling program that promotes BM sliding in neighboring uterine cells: (1) the AC expresses the transcription factor LIN-29, (2) LIN-29 activates Notch signaling and induction of the π cell fate in neighboring uterine cells (*Newman et al., 2000*), (3) as a part of π cell fate induction, Notch activation upregulates *ctg-1,* which restricts trafficking of the BM adhesion receptor DGN-1 (dystroglycan) to the cell-BM interface and allows BM sliding. Together, these observations suggest that after AC invasion, the uterine π cells loosen BM adhesion, allowing the BM to slide along the π cells to a precise site determined by the dividing vulE and F cells and non-dividing vulD. This model is supported by the observed flexibility of BM boundary position, which can be adjusted by manipulating vulval cell divisions. The distinct, but temporally coordinated, mechanisms controlling BM movement in the uterine and vulval cells precisely position the BM gap boundary to ensure robust uterine-vulval attachment.

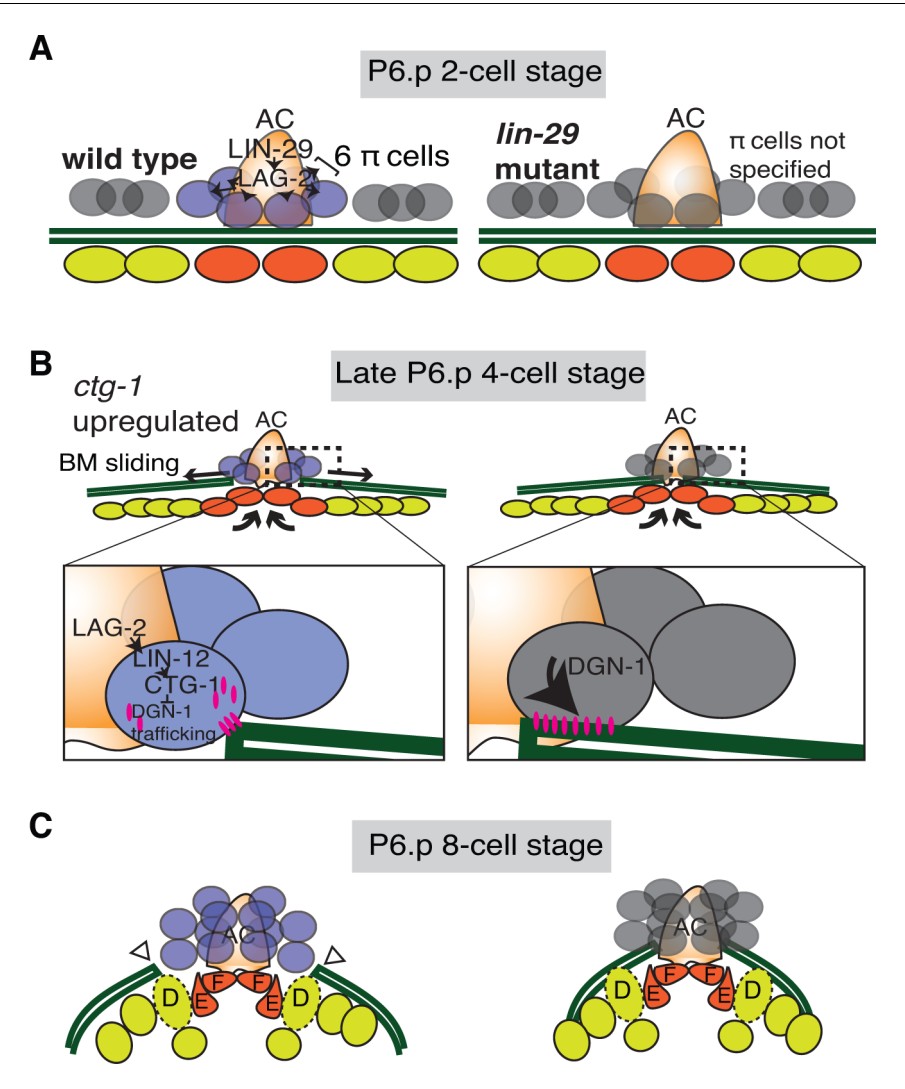

**Figure 9.** Summary of uterine boundary cell regulation of BM sliding. (**A**) A schematic diagram of the mid L3 stage in wild type (left). The Kruppel-family EGR protein LIN-29 acts in the AC to maintain LAG-2 (Notch ligand) expression, which induces the π cell fate in the six VU cells that contact the AC. This specification event fails to occur in *lin-29* mutants (right). (**B**) During the late L3 stage after AC invasion, invagination, growth, and division of the vulval cells generates forces that promote BM sliding in wild type animals (left). Within the π cells (inset), LAG-2 binding of LIN-12(Notch) activates transcriptional upregulation of *ctg-1*, which encodes a Sec14-GOLD phospholipid transfer protein. CTG-1 restricts BM receptor DGN-1 (dystroglycan) trafficking to the cell-BM interface of the π cells, thus decreasing their adhesion to the BM and allowing BM sliding. In *lin-29* mutants (right) *ctg-1* expression is not upregulated in the unspecified ventral uterine descendant cells, and more DGN-1 is trafficked to the cell-BM interface, maintaining cell adhesion to the BM and preventing sliding. (**C**) At the early-to-mid L4, the BM gap edges have slid (arrowheads) and stabilize over the vulval D cells and at the edges of uterine π cells in wild type animals (left). In contrast, the BM does not slide in *lin-29* mutants and maintains its position adjacent to the AC (right).

Very few examples of gene regulatory networks guiding morphogenetic cellular behaviors leading to cell fate specification are known (*Christiaen et al., 2008*). Using a focused RNAi screen we found that *ctg-1*, which encodes a Sec14 family phosphatidylinositol-transfer protein (PITP) (*Tripathi et al., 2014*), is a crucial target of Notch signaling during π cell fate specification that promotes BM sliding. Consistent with *ctg-1* being a direct transcriptional target of Notch, the *ctg-1* upstream regulatory region contains 19 predicted binding sites for the Notch transcriptional activator LAG-1 (Suppressor

of Hairless) (**Yoo, 2004**). Furthermore, *ctg-1* expression is upregulated in the π cells at the time of activated LIN-12 (Notch) signaling, and its expression is dependent on LIN-29-directed Notch signaling. In addition, uterine-cell specific RNAi and genetic loss of *ctg-1* reduced BM sliding, which phenocopies loss of Notch signaling in these cells. Sec14 family PITPs are unlikely to be true carriers of phospholipids. Instead, studies suggest that Sec14 family PITPs function as scaffolds or nanoreactors that present phosphatidylinositol to phosphatidylinositol-4 kinase (PI4K) to regulate specific protein and membrane trafficking pathways in the trans-Golgi and endosomal recycling system (**Cockcroft and Garner, 2011**; **Curwin, 2013**; **Curwin et al., 2009**; **Mousley et al., 2008**; **Schaaf et al., 2008**; **Tripathi et al., 2014**). Consistent with this notion, our work also indicates that the *C. elegans* PI4K, *pifk-1*, is required in the uterine cells to promote BM sliding.

The Sec14 family has expanded significantly in multicellular eukaryotes. Whereas yeast have six Sec14 family members, we found *C. elegans* has at least 16 and vertebrates have at least 20 (**Bankaitis et al., 2010**). Little, however, is known about the cellular process controlled by many these proteins (**Grabon et al., 2015**). Our work suggests that CTG-1 regulates the trafficking of DGN-1(dystroglycan) to the cell surface and promotes de-adhesion from BM to control BM sliding. Notably, localization and photobleaching studies indicate that the integrin heterodimer INA-1/PAT-3, another key BM receptor, was unaffected by loss of *ctg-1*, suggesting that CTG-1 function is highly specific. Loss of LIN-29-mediated Notch signaling (and thus *ctg-1* expression in the uterine cells) led to an increase of DGN-1 on the surface of uterine cells, but did not alter *dgn-1* gene expression. Photobleaching experiments in wild type and *lin-29* mutants (lack upregulation of *ctg-1*) indicated that *ctg-1* expression in the π cells restricts the rate of DGN-1 trafficking to the cell-BM interface. Interestingly, in CTG-1 the Sec14 domain is paired with a GOLD (Golgi Dynamics) domain, which, though its function has yet to be completely characterized, may regulate the selection of proteins being trafficked from the Golgi, thus affecting their secretion to the plasma membrane or routing to the endosome-lysosomal system (**Anantharaman and Aravind, 2002**; **Schimmoller et al., 1995**). These results suggest that CTG-1 might reduce cell-BM adhesion by regulating a vesicular pathway with specificity for DGN-1, possibly slowing DGN-1 delivery to the cell surface or directing it to the endosome-lysosome system. The function of Sec14-GOLD proteins is poorly understood, and CTG-1 does not show strong localization to any specific domain in the uterine π cells. Thus, we cannot rule out possible functions for CTG-1 in mediating increased removal of DGN-1 from the cell surface or slowing lateral plasma membrane trafficking to restrict DGN-1 at the cell-BM interface. While the specific mechanism by which CTG-1 restricts DGN-1 trafficking remains unclear, we found that reduction of DGN-1 restored BM sliding in the absence of *ctg-1,* strongly supporting the functional significance of reduced localization and trafficking of DGN-1.

The two most prominent BM receptors that transduce signals and link the cells' cytoskeleton to the BM are integrin heterodimers and dystroglycan (**Bello et al., 2015**; **Yurchenco, 2011**). Unlike mammals, which construct 24 known αβ integrin heterodimers, *C. elegans* make only two, and only one of these, αINA-1/βPAT-3, is expressed with DGN-1 (dystroglycan) in the uterine π cells (**Hagedorn et al., 2009**; **Ihara et al., 2011**). Integrin and dystroglycan are co-expressed widely, and likely have many distinct roles, such as having opposing effects on extracellular-regulated kinase (ERK) activation (**Ferletta et al., 2003**). However, how these proteins interact and coordinate functions in regulating cell-BM communication and adhesion has been challenging to elucidate as a result of tissue complexity and the large family of integrin heterodimers in vertebrates (**Bello et al., 2015**; **Nakaya et al., 2013**). Interestingly, our data suggest that INA-1/PAT-3 is not downregulated to promote BM sliding in the uterine π cells. Further, unlike loss of *dgn-1*, uterine-specific loss of *ina-1* did not rescue BM sliding in the *lin-29* mutant background. Our results indicate that DGN-1 is the primary BM adhesion receptor that controls cell-BM sliding in the uterine π cells, while integrin appears to have a function(s) independent of BM sliding in these cells. These results suggest that different BM receptors may allow cells to have diverse and concurrent signaling and adhesive interactions with BM.

While dystroglycan has been most extensively studied in its association with the dystrophin complex and muscular dystrophies, it is broadly expressed and implicated in regulating many cellular processes, including BM assembly, cell migration, axon outgrowth, retinal layering, and Schwann cell wrapping (**Bello et al., 2015**; **Moore and Winder, 2010**). Notably, dystroglycan is also required for branching morphogenesis in glandular epithelium, a process that is thought to require BM sliding (**Durbeej et al., 2001**; **Harunaga et al., 2014**). Further, it has been shown that loss of basally

localized dystroglycan during chick gastrulation promotes breakdown of BMs during the epithelial-to-mesenchymal transition that forms mesoderm (*Nakaya et al., 2011*). Thus, dystroglycan may have a fundamental role in mediating BM remodeling events during development. Importantly, loss or reduction of dystroglycan is also emerging as a common event in the progression of epithelial derived cancers, including breast, prostate, colon, cervical, and renal adenocarcinomas (*Cross et al., 2008*; *Esser et al., 2013*; *Sgambato et al., 2007*, *2003*, *2006*). As regulated loss or reduction of dystroglycan during development creates and widens BM gaps through BM loss and sliding, uncontrolled loss of dystroglyan during cancer progression may lead to the breakdown of BM barriers to facilitate tumor spread. Thus, understanding mechanisms that control dystroglycan levels at the cell surface are not only important in understanding tissue morphogenesis, but may provide new strategies to halt cancer progression.

## Materials and methods

### Worm strains

Worms were reared under standard conditions at 15°C, 20°C, or 25°C (*Brenner, 1974*). N2 Bristol strain was used as wild-type nematodes. Strains were reared and viewed at 20°C or 25°C using standard techniques. In the text and figures, we use a '>' symbol for linkages to a promoter and use a '::' symbol for linkages that fuse open reading frames (*Ziel et al., 2009*). The following alleles and transgenes were used: *qyEx439 [egl-13 > cki-1::GFP], cgEx308 [dgn-1(+), rol-6(su1066), dgn-1 > GFP], qyEx557 [fos-1 > moesinABD::mCherry, myo-2 > GFP], qyEx558 [fos-1 > moesinABD:: mCherry, myo-2 > GFP], qyEx561 [fos-1 > rab-7::GFP, dgn-1 > dgn-1::mCherry, unc-119(+)], qyIs28 [ced-10::GFP], qyIs43[pat-3 > pat-3::GFP, genomic ina-1], qyIs102 [fos-1 > rde-1; myo-2 > GFP], qyIs108 [laminin::Dendra], qyIs127 [laminin::mCherry], qyIs251 [cdh-3 > lin-29a::GFP], qyIs330[laminin::mCherry], qyIs351 [unc-62 > GFP::CAAX], qyIs361[lin-29a/b > GFP], qyIs486 [dgn-1 > dgn-1:: GFP], qyIs508 [fos-1 > ctg-1::GFP];* **LGI,** *ctg-1(qy11); rhIs2 [pat-3::GFP];* **LGII,** *lin-29(ga94), lin-29(qy1), rrf-3(pk1426);* **LGIV,** *qyIs8 [laminin::GFP];* **LGV,** *kuIs29 [egl-13::GFP(pWH17) + unc-119(+)], rde-1 (ne219);* **LGX,** *sel-12(ty11), qyIs24 [cdh-3 > mCherry::PLCδ^PH^], qyIs86 [egl-13 > GFP].* Because the protruding vulva of *qy1* homozygotes prevents male mating, the *lin-29(qy1)*-containing chromosome was balanced by the balancer *mIn1[mIs14 dpy-10(e128)]* for effective mating (*Edgley and Riddle, 2001*).

### Mutagenesis screen and mutant isolation

L4 N2 Bristol strain worms were mutagenized in 4 ml M9 buffer with N-ethyl-N-nitrosourea (ENU) at a concentration of 0.5 mM in a 15 ml conical tube. The tube was placed on the rocker for 4 hr at room temperature then subject to centrifugation to collect worms for recovery. Worms were washed 3 times with 3 ml M9 buffer and resuspended in a few drops of M9, then transferred to a plate using a glass pipette. One h later, healthy worms were transferred to new plates at a density of 2 worms/plate. Five rounds of mutagenesis were performed. F2 progeny produced from the mutagenized worms were examined for the protruding-vulva (Pvl) phenotype. The fertile Pvl mutants were then subject to microscopic examination for defects in uterine-vulval attachment. During this examination, 10 fertile Pvl mutants were identified with defects in BM remodeling during uterine-vulval attachment. Nine of these had defects in AC invasion, and one, *qy1*, had a defect in the BM hole opening after invasion, suggesting that mutants that affect BM hole opening are rarer than those that alter AC invasion. Of the nine mutants with defects in AC invasion, four were mapped further. One AC invasion mutant is putative allele of *unc-6* and another an allele other *unc-40*, genes that are known to promote AC invasion (*Ziel et al., 2009*).

### Genetic mapping

A single-nucleotide-polymorphism (SNP) based mapping strategy (*Davis et al., 2005*) was used to locate the *qy1* mutation to between 11777294 and 12029092 on Chromosome II. Briefly, three hundred F2 Pvl progeny of multiple *qy1/CB4856* F1 hermaphrodites that were produced by crossing *CB4856* males into *qy1/mIn1* were individually isolated onto each well of 24-well plates. After four days, self-progeny from these F2 progeny were washed off the plate using water (>30 worms/plate) and placed in a single well of a 96-well plate. Worms were allowed to settle to the bottom of the

wells at 4°C for 15 min (min). Excess solution was removed to leave a final volume of 90 µl/well. Plates were stored at −80°C. For mapping PCR, the plates were thawed and 1X lysis buffer (50 mM KCl, 2.5 mM MgCl$_2$, 10 mM Tris pH 8.3, 0.45% Tween 20, 0.04% gelatin, 100 µg/ml proteinase K (freshly added)) was added. Worms were lysed by incubation at 65°C, 1 hr and 95°C, 15 min to extract DNA. PCR templates were frozen at −80°C and thawed prior to each use. For each PCR, 1 µl of worm lysate was added into each well of the 96-well plate that then received 19 µl of a PCR mix containing 14 µl water, 2 µl 10X buffer, 0.4 µl 10 mM dNTP, 2 µl each primer (10 µM), and 0.2 µl Taq (5 units/µl). PCR conditions: 93°C, 2 min, 35 cycles (93°C, 20 s (s), 58°C, 30 s, 72°C, 1 min), 72°C, 5 min, 10°C for holding. The restriction enzyme DraI 0.26 µl (20 unit/µl) with its 10x buffer was added into each PCR well to make final digestion volume of 22.26 µl. Within the region identified, 4 genes were previously reported to have a Pvl phenotype: ZK930.3, lin-29, ash-2, and mcm-2, which we confirmed by RNAi. qy1 was identified as lin-29 when it failed to complement lin-29(n546).

## Photoconversion of Dendra tagged laminin

Transgenic animals expressing laminin::Dendra were photoconverted using a Zeiss LSM 510 confocal microscope (Zeiss Microimaging), equipped with a 63x objective, scanning regions of interest with a 405 nm laser at 1 mW power for 30 s. After photoconversion, images were captured using a spinning disc confocal microscope. Animals were recovered from the agar pad, left to develop on the plates with OP50 at 25°C for the specified amount of time then reimaged. Identical settings were used to acquire images at all times. Acquired images were processed using Imaris (Bitplane).

## Cell ablations

Laser-induced cell ablations of the AC and VU were performed as previously described (Bargmann, 1995). ACs were ablated at the P6.p 4-cell stage after clearing of the BM by the AC was complete. For ventral uterine (VU) cell ablations, the 2 VUs that flanked the AC on either the anterior or posterior side (Z4.aaa and Z1.ppa or Z4.aap and Z1.ppp) were ablated at the early P6.p 2-cell stage prior to π cell specification. Worms were recovered from slides to OP50 plates at 25°C and imaged at the L4 stage.

## Microscopy, image acquisition and quantitative analysis of BM opening

Images were acquired using a Zeiss AxioImager microscope with a 100x Plan-APOCHROMAT objective and equipped with a Yokogawa CSU-10 spinning disc confocal controlled by iVision (Biovision Technologies) or Micromanager software, or using a Zeiss AxioImager microscope with a 100x Plan-APOCHROMAT objective and with a Zeiss AxioCam MRm CCD camera controlled by Axiovision software (Zeiss Microimaging). Acquired images were processed using ImageJ 1.40 ImageJ 1.47d and Photoshop CS3 Extended or Photoshop CC (Adobe Systems). Fluorescence intensities were measured as Mean Gray Values using ImageJ. Three-dimensional reconstructions were built from confocal z-stacks, analyzed and exported using Imaris 7.4 (Bitplane). The diameter of the BM opening along the anterior-posterior axis was measured as the size of the opening using single-channel images of laminin::GFP, laminin::mCherry or the phase dense line of the BM seen by DIC in plane of focus for the AC using ImageJ. For CKI-1 π division arrest, examination of the number of π cell nuclei by DIC as well as GFP expression in the π cells were used to determine whether π cell division had occurred, and only animals in which all 6 π cells failed to divide were scored.

## Molecular biology and transgenic strains

Standard molecular biology and transgenic techniques were used to generate PCR fusion products (Hobert, 2002), plasmids, and transgenic animals (Mello and Fire, 1995). To generate the transcriptional reporter for the lin-29a/ b gene, the 5′ cis-regulatory element (5′ CRE) 5.4 kb upstream of the ATG start codon of the lin-29a/ b gene was amplified. The 5′ CRE sequence was then fused in frame to GFP coding sequence (PCR amplified from the vector pPD95.81) using PCR fusion. For the transcriptional reporter of the lin-29c gene, the 5′ CRE 4.2 kb upstream of lin-29c coding sequence was PCR amplified and fused in frame to GFP coding sequence (PCR amplified from the vector pPD95.81) using PCR fusion. AC-specific lin-29a::GFP was generated by fusing GFP coding sequence with the amplicon cdh-3 > lin-29a using PCR fusion. To generate the amplicon cdh-3 > lin-29a, lin-29b cDNA was first PCR amplified and subcloned into the vector pPD95.81 containing the

cdh-3 promoter (*Sherwood et al., 2005*) at XhoI and HindIII sites. The coding sequence of *cdh-3 > lin-29a* was PCR amplified from the vector *cdh-3 > lin-29b* using the New England Laboratory site-directed-mutagenesis protocol with a primer containing a sequence encoding four *lin-29a* specific amino acids. To generate the *egl-13 > cki-1::GFP* construct, the 5 kb promoter region of *egl-13* was amplified from plasmid pWH17 and fused with to CKI-1::GFP amplified from the recombineered fosmid WRM0626bF02 (wTRG5.1_2491680425634929_G10). To generate the DGN-1 translational reporter, 2.6 kb upstream of the start codon to the end of the coding sequence of *dgn-1* was amplified from N2 genomic DNA and fused in frame to the GFP coding sequence amplified from vector pPD95.75, based on a previously published construct that rescues the sterility phenotype of *dgn-1* (*Johnson et al., 2006*). To generate *fos-1* promoter containing constructs, pBlueScript containing the *fos-1* promoter (*Sherwood et al., 2005*) was digested with ApaI and BamHI, then fused with the *ctg-1* coding sequence amplified from genomic DNA and the GFP coding sequence amplified from pPD95.75 for *fos-1 > ctg-1::GFP*, the moesin::ABD coding sequence amplified from pJWZ6 for *fos-1 > moesinABD::mCherry*, and the *rab-7::GFP* coding sequence amplified from pHD69 for *fos-1 > rab-7::GFP*, using NEBuilder HiFi DNA Assembly Cloning. Transgenic worms were created by co-injection of expression constructs with the transformation marker pPD#MM016B (*unc-119*), or the co-injection marker (*myo-2 > GFP*) or both into the germline of *unc-119(ed4)* mutants or wild type worms. These markers were injected with EcoRI-digested salmon sperm DNA and pBluescript II at 50 ng/µl as carrier DNA along with the expression constructs, which were normally injected at 10–50 ng/µl. Integrated strains were generated as described previously (*Inoue et al., 2002*). To generate Y75B8A.24 RNAi, the first 800 bp of the coding sequence of the gene were amplified from genomic DNA and inserted into the vector L4440, which was digested with HindIII and KpnI.

## CRISPR-Cas9 mediated generation of *ctg-1(qy11)*

A GFP knock-in that removes the majority of the coding region of *ctg-1* was generated using the CRISPR-Cas9 system (*Dickinson et al., 2013*). Two sgRNAs with sequences 5'- AATTCGACAACTAC TATTCG-3' and 5'-AATATTGTAGCGCAATTGCT-3' were used to induce double stranded breaks in the final exon of *ctg-1*. A repair template was constructed with 1.5 kb upstream of the coding start site plus the first 20 bp of *ctg-1*, followed by GFP, *cbr-unc-119*, and 1.7 kb of sequence downstream of the *ctg-1* stop codon. Plasmids containing guide RNAs and Cas9 were co-injected at a concentration of 25 ng/µL each, and the repair template was injected at a concentration of 10 ng/µL.

## Fluorescence reovery after photobleaching (FRAP) experiments

A Zeiss LSM 510 laser scanning confocal microscope with a 100x objective was used for initial bleaching and imaging the DGN-1::GFP reporter. In the plane of focus for the AC, a 1 µm by 1 µm region of interest was defined at the edge of the BM gap or next to the boundary of the AC if no visible gap in the BM was present. Regions of interest were bleached at 488 nm for 100 iterations at 100% power. Further imaging was conducted every 30 s for 9 min. The protocol was repeated for DGN-1::GFP and PAT-3::GFP using a Nikon Ti-U with the 'Plan Apo VC' 100x objective, Yokogawa spinning disc unit, Andor iXon EM-CCD, and Andor ALC601 series laser. ROIs were bleached for 20 iterations at 10% intensity.

## RNA interference

dsRNA was delivered via feeding to synchronyzed L1 larvae. Phenotypes were subsequently scored at the early L4 stage. All RNAi experiments examining BM gap opening were conducted in a blinded manner, where the treatment was masked from person quantifying the opening. Constructs were obtained from the Vidal (*Rual et al., 2004*) and Ahringer (*Simmer et al., 2003*) libraries. The L4440 empty vector was used as a negative control. Clones targeting *lag-2, ctg-1, ina-1, dgn-1* were verified by sequencing. Screening of Notch targets was scored in *rrf-3* worms containing laminin::mCherry and the AC marker (CAAX::GFP). Uterine-specific RNAi was scored in strains containing RDE-1 expressed under the control of the *fos-1a* promoter in an *rde-1* mutant background (*Hagedorn et al., 2009*). Data was collected in 3 independent trials for uterine-specific *lag-2* RNAi and *dgn-1* RNAi in the *lin-29(qy1)* background, and 2 trials for *ctg-1* RNAi and PI4-kinase RNAi.

## F-actin quantification

Fluorescence intensities were measured as the mean gray values of 0.3- μm x 0.3- μm regions of interest selected either adjacent to the edge of the expanding BM gap or no longer in contact with the BM (past the zone of BM sliding) in the π cells at the P6.p 6-cell stage. In *lin-29(qy1)* mutants, only areas adjacent to the edge of the BM gap were measured, as regions beyond the zone of BM sliding were not visible (due to the loss of BM sliding). To account for the varied expression levels seen in extrachromosomal array lines, fluorescence intensities were normalized using the fluorescence intensity of an apical region of the same size.

## RAB-7 and DGN-1 Co-localization

RAB-7::GFP positive vesicles were selected and marked using ImageJ. Regions of interest were then overlaid on images of corresponding DGN-1::mCherry and assessed for overlap.

## Identification of *C. elegans* Sec14 family members

Textpresso homology/orthology data and Gene ontology and BLAST from the WS249 version of Wormbase were searched for hSEC14L2, SEC14, and CRAL-TRIO.

## Statistical analysis

Statistical analysis was performed using JMP version 12.0 (SAS Institute) or Microsoft Excel, using a two-tailed unpaired Student's t test, two-tailed Fisher's exact test, or nonparametric Wilcoxon rank sum test. Figure legends specify when each test was used. Sample sizes were validated a posteriori for statistical significance and variance (for parametric tests). Normality was assessed using a Shapiro-Wilk's normality test.

## Acknowledgements

Some strains were provided by the CGC, which is funded by the NIH Office of Research Infrastructure Programs (P40 OD010440). We thank L Kelley for technical advice on photobleaching experiments, K Gordon and S Payne for comments on the manuscript, and members of the Sherwood lab for suggestions throughout the course of this project. This work was supported by the Pew Scholars Program in the Biomedical Sciences and NIH grants GM079320 and GM100083 to DRS.

## Additional information

### Funding

| Funder | Grant reference number | Author |
|---|---|---|
| National Institute of General Medical Sciences | GM079320 | David R Sherwood |
| National Institute of General Medical Sciences | GM100083 | David R Sherwood |
| Pew Charitable Trusts | Pew Scholars Program in the Biomedical Sciences | David R Sherwood |

The funders had no role in study design, data collection and interpretation, or the decision to submit the work for publication.

### Author contributions

STHM, ZW, DRS, Conception and design, Acquisition of data, Analysis and interpretation of data, Drafting or revising the article; LML, AC, Conception and design, Acquisition of data; ELH, LW, WS, QC, Conception and design, Acquisition of data, Analysis and interpretation of data

### Author ORCIDs

David R Sherwood, http://orcid.org/0000-0003-2245-2334

## Additional files

**Supplementary files**
• Supplementary file 1. Presumptive Notch targets screened by RNAi for BM sliding defects.
• Supplementary file 2. Sec14 family genes in the *C. elegans* genome.

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
