## [Decision Letter]

Thank you for submitting your article "Boundary cells restrict dystroglycan trafficking to control basement membrane sliding during tissue remodeling" for consideration by *eLife*. Your article has been reviewed by three peer reviewers, including Vytas Bankaitis (Reviewer #2) and Kenneth Yamada (Reviewer #3), and the evaluation has been overseen by Janet Rossant as the Senior Editor and Reviewing Editor.

The reviewers have discussed the reviews with one another and the Reviewing Editor has drafted this decision to help you prepare a revised submission.

Summary:

In this manuscript, the authors use powerful forward genetics, gene silencing, cell ablation and imaging methods to investigate BM sliding in the worm. The question is of general interest as tissue development and organogenesis relies on accurate control of the sliding of epithelial cells along BMs, yet little is known about how the sliding process is regulated. In this work, the authors systematically investigate the BM sliding process in the context of uterine-vulval attachment. The experiments described are logical, systematic and convincingly demonstrate that the invasive uterine AC communicates with neighboring uterine cells to activate a Notch signaling pathway whose major target in the pathway is a Sec14-GOLD protein of previously unknown function. FRAP data suggest that the Sec14-GOLD protein specifically and negatively regulates dystroglycan trafficking to the cell surface. It is modulation of the BM adhesion activity of the cell surface dystroglycan pool that regulates BM sliding at the BM gap boundary.

Essential revisions:

The data leading to identification of the CTG-1 Sec14-GOLD protein as the main target of Notch signaling in the vulva-uterine BM sliding pathway are comprehensive and beautiful. Those alone make a very nice story and revisions needed are mostly minor. However, the true novelty of the work is that a biological function is described for a Sec14-GOLD protein. Although a number of Sec14 proteins are very well understood from biological and structural perspectives in yeast and in plants (a point miscast by the authors in their general discussions of the topic), the Sec14-GOLD proteins are poorly understood. Where this work could make truly unique inroads is in characterizing how CTG-1 functions. They do not take full advantage of this opportunity. The following comments are suggested to the authors for strengthening this work.

A) Regarding CTG-1, the interpretations offered in this manuscript for how this protein works largely follow those of those yeast studies. But, Sec14 and other Sec14-like proteins also stimulate the activities of the Stt4 family of PI4kinases. Is the BM phenotype the authors report for *pifk* silencing specific or is it a general phenotype of PI4K knockdown? How about PI4P 5-kinase knockdown?

B) It is not clear that the FRAP data of Figure 8 demonstrate enhanced trafficking of dystroglycan to the cell surface. That interpretation implies that CTG-1 is regulating a cargo-specific trafficking pathway to the plasma membrane. While certainly possible, the authors need to repeat the experiment with another cargo not involved in the process (Ina1, Pat2?) in the FRAP experiment as control.

C) As follow-up to (B), it seems the FRAP experiments do not exclude a role for CTG-1 in dystroglycan endocytosis. The dystroglycan FRAP data are consistent with more exocytosis or just more rapid recycling or more lateral diffusion – these need to be considered.

Cargo-specific endocytic systems are at least as likely a mechanism as cargo-specific forward trafficking pathways. In that regard, what effect do CTG-1 mutations have on the actin cytoskeleton? The phosphoinositide-actin-endocytosis axis is an attractive one to consider.

If you can add experimental data to address more completely the function of CTG-1, it would strengthen the impact of the paper. However, this may take longer than the two months that *eLife* allows for revisions. The reviewers are generally positive about the paper and so would be prepared to re-review a revised manuscript that addresses some of the issues.

[Editors' note: further revisions were requested prior to acceptance, as described below.]

Thank you for resubmitting your work entitled "Boundary cells restrict dystroglycan trafficking to control basement membrane sliding during tissue remodeling" for further consideration at *eLife*. Your revised article has been favorably evaluated by Janet Rossant as the Senior Editor and Reviewing Editor, and two reviewers.

Both reviewers are happy with your revisions and the paper is almost ready for acceptance. There are some remaining issues that need to be addressed before acceptance, as outlined below:

1) There are two concerns with Figure 2. The first is that it is quite pixelated, even on the original eps file; can the authors find a better-quality substitute? The second, much more significant concern is that upon further examination, the left bracket for wild type may show an incorrectly wide gap. Although the laminin grayscale image is not shown, the overlay shows white or gray at the upper right rather than green, which means that the magenta (laminin) is co-localizing with the green. Consequently, the right bracket should be significantly further to the left. If the authors disagree, they could resolve this type of concern by showing data from the laminin-staining channel by itself. In that regard, for their quantifications, did the authors measure such single-channel data as opposed to this type of less-definitive merged image?

2) In the last paragraph of the subsection “LIN-29/AC-mediated uterine π cell fate specification is required for BM sliding”: "likely do to" should read "likely due to".

---

## [Author Response]

Essential revisions:

*The data leading to identification of the CTG-1 Sec14-GOLD protein as the main target of Notch signaling in the vulva-uterine BM sliding pathway are comprehensive and beautiful. Those alone make a very nice story and revisions needed are mostly minor. However, the true novelty of the work is that a biological function is described for a Sec14-GOLD protein. Although a number of Sec14 proteins are very well understood from biological and structural perspectives in yeast and in plants (a point miscast by the authors in their general discussions of the topic), the Sec14-GOLD proteins are poorly understood. Where this work could make truly unique inroads is in characterizing how CTG-1 functions. They do not take full advantage of this opportunity.*

To address the important point about *ctg-1* encoding a Sec14-GOLD protein (a distinction we only eluded to in the Discussion) in the Abstract, Results section and Discussion we have more specifically delineated this point, including annotating how many Sec14 family members in *C. elegans* contain a GOLD domain ([Supplementary-material SD23-data]).

*The following comments are suggested to the authors for strengthening this work.*

*A) Regarding CTG-1, the interpretations offered in this manuscript for how this protein works largely follow those of those yeast studies. But, Sec14 and other Sec14-like proteins also stimulate the activities of the Stt4 family of PI4kinases. Is the BM phenotype the authors report for pifk silencing specific or is it a general phenotype of PI4K knockdown? How about PI4P 5-kinase knockdown?*

We thank the reviewers for these essential revisions and agree that we need to expand our analysis of CTG-1 function in relation to PIP4 kinases and PIP species in general. In our submitted work, we used blind analysis of basement membrane (BM) sliding (see below) to examine RNAi knockdown (in a uterine-specific RNAi sensitive strain) targeting three PIP4-kinases--*pifk-1* (ortholog to *Four wheel drive (Fwd)* in *Drosophila*, yeast Pik1, and vertebrate PI4KIIIβ), C56A3.8 (vertebrate PI4KII⟨/®) and ZC8.8 (vertebrate PI4KIIα/β).

To address these essential revisions, we made a new RNAi clone that targeted the *C. elegans* gene Y75B8A.24, which blast searches identified as the *C. elegans* Stt4 PIP4K ortholog. We also used an RNAi vector from our RNAi library (that was sequenced for verification) targeting *ppk-1*, the only *C. elegans* PI5K (Weinkove et al., 2008, Dev Biol 384-97). RNAi targeting *ppk-1* should reduce PI(4,5)P_2_ production. In blind analysis, loss of both C56A3.8 and *ppk-1* in the uterine cells did not alter BM sliding, suggesting that neither is a downstream effector of CTG-1 function in the uterine π cells. This new data is presented in Figure 7—figure supplement 3. We have also added the following text to the paper:

To the Results section entitled, “LIN-29/CTG-1 reduces dystroglycan localization and trafficking in the π cells”at the end of paragraph one:

“To determine whether CTG-1 might regulate BM sliding through a PI4-kinase, we examined uterine-cell specific RNAi-mediated knockdown of the four PI4-kinases encoded in the *C. elegans* genome. […] As Sec14 and Pik1/ PI4KIIIβ are implicated as regulators of vesicle trafficking (Clayton et al., 2013; Sciorra et al., 2005), we hypothesized that *ctg-1* might be upregulated in the π cells to alter the trafficking and localization of BM adhesion receptor proteins to promote BM sliding.”

We also included the orthologs of the PI4-kinases that we examined in the figure legend of Figure 7—figure supplement 3 as follows:

“(A) Quantification of the BM gap size after treatment with L4440 control and uterine-specific *pifk-1* (yeast Pik1 and vertebrate PI4KIIIβ) RNAi, C56A3.8 (vertebrate PI4KIIα/β) RNAi, and ZC8.6 (vertebrate PI4KIIα/β) RNAi (n = 34 each), (B) Quantification of the BM gap size after treatment with L4440 control and uterine-specific Y75B8A.24 (vertebrate PI4KIIIα) (n = 34 each) RNAi, and (C) Quantification of the BM gap size after treatment with L4440 control and uterine-specific *ppk-1* (vertebrate type I PIP 5-Kinase) RNAi (n = 40 each) revealed that only *pifk-1* is required for BM sliding. (*** indicates P < 0.001 Wilcoxon rank sum test).”

*B) It is not clear that the FRAP data of Figure 8 demonstrate enhanced trafficking of dystroglycan to the cell surface. That interpretation implies that CTG-1 is regulating a cargo-specific trafficking pathway to the plasma membrane. While certainly possible, the authors need to repeat the experiment with another cargo not involved in the process (Ina1, Pat2?) in the FRAP experiment as control.*

We agree with the reviewers about the importance of testing whether the FRAP data in Figure 8 indicates a specific role for CTG-1 in dystroglycan trafficking at the cell surface. We actually performed the reviewers’ suggested experiment for FRAP on the integrin heterodimer PAT-3/INA-1 (one of two integrins in *C. elegans* and the only integrin heterodimer expressed in the π cells) using the functional expression reporter PAT-3::GFP/INA-1 (see Figure 8—figure supplement 1). We clearly did not do an adequate job of bringing our reviewers/readers attention to this important experiment. We have thus added the following to the paper to make this experiment more prominent:

We added to the ending of the second paragraph in the Results section entitled, “LIN-29/CTG-1 reduces dystroglycan localization and trafficking in the π cells”:

“These experiments indicate that CTG-1 does not regulate INA-1/PAT-3 localization or trafficking in the π cells to promote BM sliding.”

To emphasize the specific nature of CTG-1 in regulating DGN-1 trafficking, we have also added the following to the last sentence of the third paragraph of this section of the Results:

“These results suggest that LIN-29 activity in the AC leads to LAG-2/LIN-12 (Notch)-mediated upregulation of ctg-1 expression in the uterine π cells that specifically restricts the cell surface trafficking and accumulation of the BM receptor DGN-1.”

To further emphasize the FRAP data, in the Discussion section, fourth paragraph, we also added the following underlined statement:

“Notably, localization and photobleaching studies indicated that the levels and trafficking of the integrin heterodimer INA-1/PAT-3, another key BM receptor, was unaffected by loss of *ctg-1*.”

*C) As follow-up to (B), it seems the FRAP experiments do not exclude a role for CTG-1 in dystroglycan endocytosis. The dystroglycan FRAP data are consistent with more exocytosis or just more rapid recycling or more lateral diffusion – these need to be considered.*

*Cargo-specific endocytic systems are at least as likely a mechanism as cargo-specific forward trafficking pathways. In that regard, what effect do CTG-1 mutations have on the actin cytoskeleton? The phosphoinositide-actin-endocytosis axis is an attractive one to consider.*

This is a terrific point the reviewers bring up. In particular examining and broadening the possible mechanisms by which CTG-1 might restrict dystroglycan levels and trafficking at the cell surface is important to explore and acknowledge. This is particularly important, as the functions of Sec14-GOLD proteins are poorly understood. We have now examined the co-localization of DGN-1 with Rab-7, a marker for the late endosome, and found no significant difference in the number of DGN-1 containing late endosomes. In addition, we have examined the localization of F-actin in the uterine π cells and discovered that there is less F-actin in *lin-29* mutants (– *ctg-1*) at the cell-BM interface compared with wild type animals. As cortical F-actin is thought to generally act as a barrier to secretion (Eitzen G., (2003) Biochimica et Biophysica Acta 1641:175– 181), this observation is consistent with the idea that more DGN-1 may be delivered to the cell surface in *lin-29* mutants. However, reduced cortical F-actin might also promote more rapid lateral diffusion of DGN-1. Given the specificity of CTG-1 function in regulating DGN-1 trafficking (and not integrin) our data support the idea of a specific trafficking mechanism regulated by CTG-1 (either endocytosis, secretion or possibly lateral mobility). We have incorporated the new data into the manuscript and broadened our discussion on the possible function of CTG-1 in regulating DGN-1 trafficking as follows:

To account for the possibility that CTG-1 restricts lateral mobility, in the Abstractwe have changed:

“our results suggest that CTG-1 restricts BM adhesion receptor DGN-1 (dystroglycan) trafficking to the cell surface, which promotes BM sliding.”

To:

“our results suggest that CTG-1 restricts BM adhesion receptor DGN-1 (dystroglycan) trafficking to the cell-BM interface, which promotes BM sliding.”

In the Results section, “LIN-29/CTG-1 reduces dystroglycan localization and trafficking in the π cells”at the end of paragraph one:

“FRAP experiments indicated that DGN-1 recovered more quickly after photobleaching in *lin-29* mutants versus wild type animals, which may be responsible for the increased DGN-1::GFP localization (Figure 8).[…] Importantly, these observations do not rule out the possibility that CTG-1 enhances endocytic removal from the cell surface at levels that we were unable to detect or that it regulates the membrane mobility of DGN-1 to limit DGN-1 at the cell-BM interface.”

In the Discussion section, paragraph four, we also added:

“Photobleaching experiments in wild type and *lin-29* mutants (which lack upregulation of *ctg-1*) indicated that *ctg-1* expression in the π cells restricts the rate of DGN-1 trafficking to the cell-BM interface. […]While the specific mechanism by which CTG-1 restricts DGN-1 trafficking remains unclear, we found that reduction of DGN-1 restored BM sliding in the absence of *ctg-1* strongly supporting the functional significance reduced localization and trafficking of DGN-1.”

In the Materials and methods section we have added the following to describe F-actin and Rab-7/DGN-1 quantification:

“F-actin quantification

Fluorescence intensities were measured as the mean gray values of 0.3- µm x 0.3- µm regions of interest selected either adjacent to the edge of the expanding BM gap or no longer in contact with the BM (past the zone of BM sliding) in the π cells at the P6.p 6-cell stage. […] RAB-7 and DGN-1 Co-localization

RAB-7::GFP positive vesicles were selected and marked using ImageJ. Regions of interest were then overlaid on images of corresponding DGN-1::mCherry and assessed for overlap.

[Editors' note: further revisions were requested prior to acceptance, as described below.]

*1) There are two concerns with Figure 2. The first is that it is quite pixelated, even on the original eps file; can the authors find a better-quality substitute? The second, much more significant concern is that upon further examination, the left bracket for wild type may show an incorrectly wide gap. Although the laminin grayscale image is not shown, the overlay shows white or gray at the upper right rather than green, which means that the magenta (laminin) is co-localizing with the green. Consequently, the right bracket should be significantly further to the left. If the authors disagree, they could resolve this type of concern by showing data from the laminin-staining channel by itself. In that regard, for their quantifications, did the authors measure such single-channel data as opposed to this type of less-definitive merged image?*

First, we agree with the reviewers that the image presented in our manuscript is pixelated. Because the signal for laminin::mCherry is weak, images were acquired using pixel binning to maximize intensity. However, this led to a reduction in overall image quality. We have replaced the images in this figure with ones acquired under imaging conditions without binning using longer exposure times to resolve the pixilation issue.

Second, we thank the reviewers for their careful attention to our data pinpointing the boundaries of the BM gap. The reviewer has astutely noted that an area of signal exists (dashed arrow in the laminin::GFP grayscale Figure 10) to the left of the BM gap edge that we have identified (solid arrow). This isolated puncta of BM is a small piece of BM that was internalized during AC invasion and not the boundary of the BM that has slid. AC invasion involves physical forces that move and break up this barrier (Hagedorn et al., (2013) Journal of Cell Biol., 10; 201(6): 903-13) resulting in small portions of BM that are internalized or remain outside of the AC after invasion. We often see small puncta of isolated BM after AC invasion (for example see Figure 1 in Morrissey et al., PLOS Genetics, Feb 29; 12(2); e1005905 and Figure 1 (internalized) and 1F (outside the AC) in Ihara et al., (2011) Nat Cell Biol. 13:641-651).

It is also important to point out that we have extensively characterized BM sliding and stabilization during uterine-vulval attachment and found that in wild type worms, as presented in the image in question, the BM gap edge always stabilizes over vulD (Ihara et al., (2011) Nat Cell Biol. 13:641-651 and Matus et al., (2011)Nat Commun. 5: 4184). In addition, the BM gap boundary displays increased laminin relative to the rest of the BM (Matus et al., (2011)Nat Commun. 5: 4184, Figure 6). Neither the positioning, nor the relative intensity of the puncta in question is consistent with these characteristics of the BM gap edge.

We agree that measurement of merged, multi-channel images may obscure the edges of the BM gap and wish to reassure the reviewers that measurements were made on single-channel images for quantifications.

We have made the following changes to address these concerns:

We have replaced the images of vulval precursor morphology in wild type and *lin-29* mutants in Figure 2 with higher quality (i.e., not pixelated) images.

We are presenting the grayscale image for the laminin::mCherry channel for the image appearing in Figure 2 in the manuscript previously submitted on August 23, 2016 for further examination here (see Figure 10):

Author response image 1.(**A**) and (**B**) Are the image panels from the manuscript submitted on August 23, 2016.(**C**) The single-channel image of laminin::mCherry showing the BM gap edge (solid arrow) and a puncta of laminin signal to the left (dotted arrow) that is discontinuous from the rest of the BM. D) A heatmap of the laminin::mCherry fluorescence showing an increase in laminin at the previously identified BM gap edge (arrowhead).**DOI:**
http://dx.doi.org/10.7554/eLife.17218.044

In addition, we have included the single-channel images of the laminin::mCherry for the new images in Figure 2. Upon close examination of the new wild type image included in Figure 2, there is also a small puncta of internalized laminin::mCherry, similar to the one noted by the reviewer in our previously submitted image. We have amended the legend for Figure 2, explaining that the presence of this laminin signal does not represent the BM gap boundary.

from:

“(B) Vulval morphology in wild type (left) versus a *qy1* mutant (right). The number (see text) and positions of vulF, vulE, and vulD (outlined in CAAX::GFP) were normal in *lin-29(qy1)* animals (top panels, grayscale). Overlay of VPCs (green) with laminin::mCherry (magenta; bottom) reveals boundaries of the narrower BM gap in the *lin-29(qy1)* mutant (bracket). All images lateral, central plane. Scale Bars, 5µm.”

to:

“(B) Vulval morphology in wild type (left) versus a *qy1* mutant (right). The number (see text) and positions of vulF, vulE, and vulD (outlined in CAAX::GFP) were normal in *lin-29(qy1)* animals (top panels: grayscale, bottom panels: green), while the BM (laminin::mCherry, middle panels: grayscale and bottom panels: magenta) gap boundaries are narrower in the *lin-29(qy1)* mutant. (bracket). The arrow in the middle panel for the wild type denotes a puncta oflaminin internalized by the AC during invasion, and does not represent the edge of the BM gap (which is located above vulD in wild type and is characterized by a build-up of laminin). All images lateral, central plane. Scale Bars, 5µm.”

We have added text to the Methods section describing the measurement of the BM gap opening to clarify that we used single-channel images for the quantification:

“The diameter of the BM opening along the anterior-posterior axis was measured as the size of the opening using single-channel images of laminin::GFP, laminin::mCherry or the phase dense line of the BM seen by DIC in plane of focus for the AC using ImageJ.”

2) In the last paragraph of the subsection “LIN-29/AC-mediated uterine π cell fate specification is required for BM sliding”: "likely do to" should read "likely due to"

We thank the reviewers for their attention to detail in noting this error. It has now been corrected in the manuscript.